# BIGFix: Bidirectional Image Generation with Token Fixing

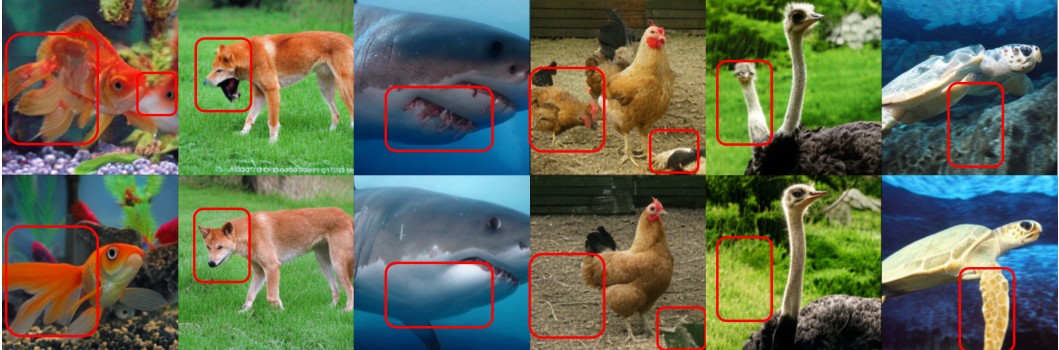

Figure 1: **Self-correction:** After the first 6 unmasking steps to create the overall structure, we proceed with two generations: without correction (top) and ours with correction (bottom). Correction solves structural errors such as supernumerary or missing elements.

## Abstract

Recent advances in image and video generation have raised significant interest from both academia and industry. A key challenge in this field is to improve both inference efficiency and quality, as model size and the number of inference steps directly impact the commercial viability of generative models while also posing fundamental scientific challenges. A promising direction involves combining auto-regressive sequential token modeling with multi-token prediction per step, reducing inference time by up to an order of magnitude. However, predicting multiple tokens in parallel can introduce structural inconsistencies due to token incompatibilities, as capturing complex joint dependencies during training remains challenging. Traditionally, once tokens are sampled, there is no mechanism to backtrack and refine erroneous predictions. We propose a method for self-correcting image generation by iteratively refining sampled tokens. We achieve this with a novel training scheme that injects random tokens in the context, improving robustness and enabling token fixing during sampling. Our method preserves the efficiency benefits of parallel token prediction while significantly enhancing generation quality. We evaluate our approach on class-to-image (ImageNet-256, CIFAR-10), text-to-image (cc12m, Laion), class-to-video (UCF-101) and image-to-video (NuScenes), demonstrating substantial improvements across multiple tasks.

## 1 Introduction

Surrounded by the aroma of freshly brewed coffee, casual chat in the office shifts from everyday topics to the latest breakthroughs in image generation. As usual, the discussion quickly shifts towards the constant push for state-of-the-art models in computer vision, reflecting the rapid evolution in the field and the drive to refine its techniques.

**Eritos**[1]: "Have you seen the news from BlackForest (Labs, 2024)? The quality is mesmerizing, indistinguishable from reality! Far superior to those chicken images you've been generating for months."

**Theoros**[2]: "Indeed, I saw it! They used flow matching, scaling it across many images, parameters, and GPUs. It's impressive how they managed to push the boundaries. But, for the record, my generated chickens are also quite remarkable! (Figure 12b)"

**E**: "Flow matching? Again a new framework? Is it related to diffusion model?"

**T**: "Flow matching (Lipman et al., 2023) is a method that teaches the model to transform a simple distribution, like a Gaussian, into a complex one, such as an image, by following smooth, continuous paths. Flow matching and diffusion (Ho et al., 2020) currently lead image generation, surpassing traditional methods like GAN (Goodfellow et al., 2014) or VAE (Kingma & Welling, 2014)."

**E**: "I thought that auto-regressive model, like GPT (Radford et al., 2019), was SOTA for data generation. I know text is discrete but an image is also a collection of discrete pixel values from 0 to 255, right? Why can't you just generate a pixel like you generate a word?"

**T**: "Technically, you can (Chen et al., 2020), but only for very small images. And for a $256 \times 256$ image, you'd need 65,536 iterations to generate just one sample... and that's assuming a single RGB value per pixel. Then you have three color channels, so the vocabulary size is $256^3$. Imagine the number of parameters and the compute power needed for training and inference. It's a nightmare."

**E**: "Indeed. But scaling usually works (Hoffmann et al., 2022), right?"

**T**: "Sure, but that's not all. Images are inherently 2D structures, yet auto-regressive methods (Sun et al., 2024) often enforce a 1D sequential representation that ignores spatial organization. The notion of geometric neighborhood is very meaningful for pixels. And this is different in text, where semantics and positional distance are much less correlated (Tian et al., 2024)."

**E**: "Okay, but what about compressing both the image resolution and its range of values?"

**T**: "Well, yes. Techniques using vector quantization (Esser et al., 2020) tokenize images, reducing their resolution and converting them into a discrete set of tokens (Mentzer et al., 2024; Yu et al., 2024). But synthesizing images token-by-token remains computationally expensive even when predicting in the latent space. Moreover, causal attention, which works well for text, is not ideal for images, as a basic raster-scan order fails to capture the conditional structure of images."

**E**: "So, what's the alternative? Can't we generate tokens in a different order?"

**T**: "This brings us to Masked Generative Image Transformer (Chang et al., 2022; 2023)! These models are trained with a reconstruction objective to predict all tokens at once. During inference, the model progressively fills a fully-masked image by predicting multiple tokens in parallel without fixing the order. This framework accelerates the image generation process, producing results in just a few steps."

**E**: "Amazing! Then why isn't everyone using this?"

**T**: "There are couple of challenges. MaskGIT iteratively reveals image tokens, which can lead to sampling discrepancies when incompatible tokens are independently sampled simultaneously (Lezama et al., 2022). Moreover, unlike diffusion models, MaskGIT cannot correct previous mistakes, once a sample is generated, it stays forever. Unless someone finds a way to fix this ... "

**T**: "Interesting. Sounds like a problem worth solving, right?!"

**Contributions.** In this paper, we propose a novel training scheme for image and video generation. Based on a multi-token prediction framework, we unlock self-correction during sampling as follows: **(1)** During training, we inject random tokens, sampled from the image distribution, into the context tokens. The network is then trained to predict both the next token in the sequence and to correct the randomly injected tokens in the context. **(2)** During sampling, the model iteratively predicts multiple

---

[1]fictive name (from erōtáō, "to ask")

[2]fictive name (from theōrein, "to seek")

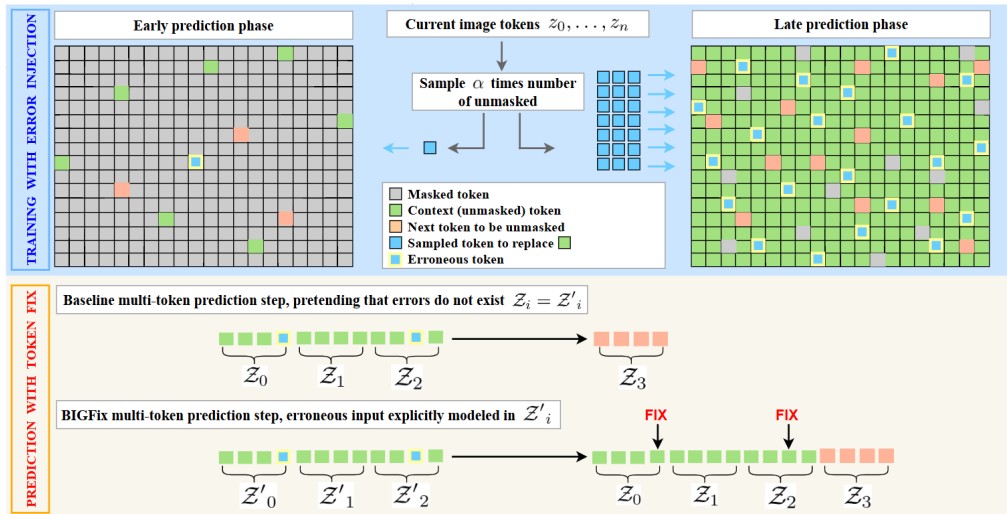

Figure 2: **Schematics.** During training (top, in blue), we perturb the input tokens by replacing a fraction $\alpha$ of the unmasked context tokens with randomly sampled ground-truth tokens from the same image. The random tokens originate from the correct image distribution but are arranged in an incorrect position. This augmentation not only enhances the model's robustness to such errors during sampling but also enables the identification and correction of incompatible tokens during inference (bottom, in orange). While baseline methods must assume the input tokens are correct and unchangeable, BIGFix allows to refine its predictions iteratively.

tokens in parallel. No random tokens are injected into the context; instead, the model is allowed to 'backtrack' and refine previously sampled tokens that exhibit structural inconsistencies Figure 2.

## 2 RELATED WORK

**Continuous Visual Generative Modeling.** Generative image modeling has long intrigued researchers, and Generative Adversarial Networks (GANs) (Goodfellow et al., 2014; Kang et al., 2023) have pioneered this effort, but suffer from mode collapse issues, which prevent them from covering the full distribution of the data (Liu et al., 2023). Diffusion Models have emerged as leading approach in generative modeling (Song et al., 2021; Dhariwal & Nichol, 2021). They are trained in two phases: noise is progressively added to images, and a model is learned to reverse this process via denoising. Since their introduction, diffusion models have achieved strong results in both text-to-image (Ramesh et al., 2022; Rombach et al., 2022) and text-to-video (Ho et al., 2022; Singer et al., 2022) generation. Building on Diffusion, Flow matching (Lipman et al., 2023) offers an alternative formulation. Using continuous normalizing flows, it efficiently transforms noise into data distributions. It has since been successfully applied to both videos (Jin et al., 2024) and images (Esser et al., 2024).

**Auto-Regressive Visual Modeling.** Inspired by advances in natural language processing (Radford et al., 2019; Devlin et al., 2019), auto-regressive and masked generative methods have been adapted for image and video synthesis. iGPT (Chen et al., 2020) modeled images as pixel sequences autoregressively, though quadratic scaling of Transformers limited its use to small images only. To overcome this issue, VQ-GAN (Esser et al., 2020) employs a vector-quantized variational autoencoder(Van Den Oord et al., 2017) with perceptual and adversarial losses, compressing images into a grid of discrete tokens for realistic reconstruction. ViT-VQGAN(Yu et al., 2022a) increases scalability with Vision Transformers, while VideoGPT (Yan et al., 2021) extends the approach to video generation. DALL-E, Parti and LlamaGen (Ramesh et al., 2021; Yu et al., 2022b; Sun et al., 2024) proposed further adaptations for the text-to-image synthesis.

**Masked Visual Modeling.** Inspired by BERT (Devlin et al., 2019), MaskGIT (Chang et al., 2022; 2023) introduced an alternative to auto-regressive modeling. Challenging the slow, row-wise to-

ken generation of auto-regressive methods, MaskGIT adopts bidirectional prediction. Trained to uncover randomly masked image tokens, it begins inference with a fully masked image, iteratively unmasking tokens to produce a full image. At each step, MaskGIT predicts all tokens but retains only the most confident ones while the remaining stay masked. MaskGIT demonstrates decent quality of generated images and up to an order of magnitude faster inference over auto-regressive models (Villegas et al., 2022; Yu et al., 2023; 2024). However, parallel token sampling overlooks inter-token dependencies, potentially yielding suboptimal results, prompting alternative sampling techniques (Lezama et al., 2022; Lee et al., 2023; Besnier et al., 2025) to address this limitation.

**Multi-token Auto-regressive Methods.** To address the limitations of auto-regressive and masked generative modeling, Visual AutoRegressive Modeling (VAR) (Tian et al., 2024) introduces a coarse-to-fine next-scale prediction strategy, auto-regressively modeling multi-scale token maps from low to high detail, while still leveraging parallel token prediction at each scale. Another line of work includes methods such as ZIPAR He et al. (2025a) and NAR (He et al., 2025b), which accelerate generation by decoding local windows of neighbor tokens in parallel, achieving substantial speedups without significantly compromising image quality.

**Auxiliary Training Strategies.** In this study, we exclude auxiliary losses such as REPA (Yu et al., 2025a), or reinforcement learning finetuning (Wallace et al., 2024) which improve the quality of the generator. Same for model distillation training (Salimans & Ho, 2022) which reduces the number of steps. While effective, and applicable in tandem with our method, they typically require either additional pre-trained networks during inference or a separate fine-tuning stage on top of the vanilla approach.

Here, we want to keep the advantage of fast inference time and to combine multi-token and auto-regressive approaches. Therefore our baseline method is Besnier et al. (2025). In contrast to the above mentioned methods, we equip the model with the ability to recover from erroneous token predictions at each prediction step. Our random token injection at training time and ability to fix errors at inference time is well suited for any multi-token prediction technique and may be easily incorporated into these frameworks.

## 3 METHOD

### 3.1 PRELIMINARIES: MULTI-TOKEN PREDICTION

**Tokenizing Images.** We represent each image $x \in \mathbb{R}^{H \times W \times 3}$ as a set of discrete tokens $\mathcal{Z} = \{z_0, z_1, \ldots, z_n\}$ using a pre-trained tokenizer encoder Enc:

$$\mathcal{Z} = \text{Enc}(x), \quad \mathcal{Z} \in C^{h \times w}$$

where $C$ denotes the token vocabulary, i.e., the possible values each $z_i$ can take, and $h \times w$ represents the spatial dimensions of the token grid. The main objective in image synthesis is to learn how to sample from the data distribution $P(\mathcal{Z})$. For convenience, we can be factorize $P(\mathcal{Z})$ as the joint distribution over all tokens:

$$P(\mathcal{Z}) = P(z_0, \ldots, z_n). \tag{1}$$

**Auto-regressive Modeling.** A powerful tool to learn the data distribution, heavily used to train LLMs, is to represent the data as a sequence and learn the probability of the next token given the previous ones (the context). Auto-regressive approaches estimate the conditional probability of each token based on the previously generated tokens:

$$P_\theta(z_i \mid z_0, z_1, \ldots, z_{i-1}), \tag{2}$$

where $\theta$ represents the model parameters. During generation, auto-regressive methods sequentially sample each token by leveraging the product of these probabilities:

$$P(z_0, \ldots, z_n) = \prod_{i=0}^{n} P(z_i \mid z_0, z_1, \ldots, z_{i-1}). \tag{3}$$

Here, $n = h \times w$ is the total number of tokens. By construction, auto-regressive models enforce sequential token generation, preventing the simultaneous sampling of multiple tokens. This constraint results in a computational bottleneck, requiring $n$ forward passes, which is prohibitively expensive for images (e.g., 1024 steps for a small $32 \times 32$ image). Furthermore, this approach forces to flatten the 2D spatial structure of image tokens into a 1D sequence, typically following a raster scan ordering. This transformation corrupts the inherent spatial relationships of tokens (See Appendix E).

**Multi-token prediction.** To mitigate the inefficiency of auto-regressive methods, we redefine the image representation as an ordered sequence of groups of tokens $\mathcal{S} = \{\mathcal{Z}_0, \mathcal{Z}_1, \ldots, \mathcal{Z}_m\}$, where each subset $\mathcal{Z}_i$ is a non-empty set of tokens selected from $\{z_i\}_{i=0}^n$. All subsets $\mathcal{Z}_s$ form a non-overlapping and complete partitioning of $\{z_i\}_{i=0}^n$. Instead of predicting one token at a time, we model the conditional probability of entire token groups $P_\theta(\mathcal{Z}_s \mid \mathcal{Z}_0, \mathcal{Z}_1, \ldots, \mathcal{Z}_{s-1})$ resulting in the factorized probability distribution $P(\mathcal{S})$:

$$P(\mathcal{Z}_0, \mathcal{Z}_1, \ldots, \mathcal{Z}_m) = \prod_{s=0}^m P(\mathcal{Z}_s \mid \mathcal{Z}_0, \mathcal{Z}_1, \ldots, \mathcal{Z}_{s-1}). \tag{4}$$

Setting $m \ll n$, i.e., reducing the number of sampling steps, significantly accelerates the generation process. Each step $s$ predicts a set of next tokens $\mathcal{Z}_s$. Moreover, the model can still be trained using the standard cross-entropy loss.

### 3.2 Toward Self-Correcting Token Generation

**On token dependencies.** However, a major drawback of sampling multiple tokens simultaneously is its sensitivity to errors. Specifically, when sampling multiple tokens in parallel, the network estimates their joint distribution under the assumption of independence (due to the cross-entropy loss):

$$P(z_a, z_b) \approx P(z_a)P(z_b),$$

where $P(z_a, z_b)$ represents the true joint probability of tokens $\{z_a, z_b\}$, and $P(z_a)P(z_b)$ is the product of their independent probabilities for any $a, b \in \{0, \ldots, n\}$. In reality, token dependencies in images often cause situations where individual tokens may have high probabilities, yet their joint probability is significantly lower, leading to unrealistic or inconsistent generations[3] (Besnier et al., 2025).

In reality, the model outputs a probability distribution for each token, and those are not independent. However, *conditioned on the context tokens*, sampling the next tokens is done independently. This assumption is sufficient to create compositional problems, which we demonstrate quantitatively and qualitatively in the next section.

**Random token injection.** To mitigate this issue and allow the network to correct mistakes arising from incompatible tokens, we inject random tokens $z_i$ (sampled from the current image distribution) in the context during training. This enables the model to learn that some context tokens may contain errors and, consequently, develop the ability to correct them. Among all context tokens in $\{z_0, z_1, \ldots, z_{i-1}\}$, we randomly replace some of them with random tokens sampled from the clean image distribution $P(\mathcal{Z})$, thereby producing a second corrupted token representation $\mathcal{Z}'$, which, similarly to $\mathcal{Z}$, is a set of tokens taken from $\{z_i\}_{i=0}^n$ with the difference that some $z_i$ may be sampled multiple times, creating the corruption. Then, $\mathcal{Z}'$ is a set of $z_j$ tokens sampled accordingly:

$$\forall j \in \{0, \ldots, i-1\}, z_j' = \begin{cases} z_j, & \text{if } u \sim \mathcal{U}(0,1) > \alpha, \\ z_* \sim P(\mathcal{Z}), & \text{otherwise,} \end{cases} \tag{5}$$

where $z_*$ is a random token sampled from the distribution of the same clean tokenized image $\mathcal{Z}$.

---

[3]For example, in a picture, a person's head might appear in two different locations. While each position may have a high likelihood for the token corresponding to the 'head' independently, selecting both simultaneously could result in a person with two heads—an improbable and unrealistic outcome. Such errors may propagate through the sampling process, leading to inconsistencies in predictions.

**Model Training.** We inject random token and train our model to correct them by maximizing the likelihood of $P_\theta(\mathcal{Z}_0, \ldots, \mathcal{Z}_k \mid \mathcal{Z}'_0, \ldots, \mathcal{Z}'_k)$ which corresponds to minimizing :

$$\mathcal{L}_{context}(\theta) = -\mathbb{E}_{\mathcal{Z}, \mathcal{Z}'} \left[ \sum_{k=0}^{s-1} \log P_\theta(\mathcal{Z}_0, \ldots, \mathcal{Z}_k \mid \mathcal{Z}'_0, \ldots, \mathcal{Z}'_k) \right]. \qquad (6)$$

where $k < s$ ensures that we inject random tokens only in the context.

Moreover, we train the model to predict the next clean token group $\mathcal{Z}_s$ given a sequence of noisy token groups $\{\mathcal{Z}'_0, \ldots, \mathcal{Z}'_{s-1}\}$, thereby making it a generative model. We minimize the expected negative log-likelihood with respect to $\theta$:

$$\mathcal{L}_{next}(\theta) = -\mathbb{E}_{\mathcal{Z}, \mathcal{Z}'} \left[ \sum_{s=0}^{m} \log P_\theta(\mathcal{Z}_s \mid \mathcal{Z}'_0, \mathcal{Z}'_1, \ldots, \mathcal{Z}'_{s-1}) \right]. \qquad (7)$$

In practice, we train the model to minimize the sum $\mathcal{L}(\theta) = \mathcal{L}_{next}(\theta) + \mathcal{L}_{context}(\theta)$. During sampling, no injections are made into the context; instead, we allow the model to iteratively correct its previous steps.

**Bidirectional Halton ordering** Building on prior work (Besnier et al., 2025), which serves as our main baseline method, we construct the set $\mathcal{S} = \{\mathcal{Z}_0, \mathcal{Z}_1, \ldots, \mathcal{Z}_m\}$ using the Halton sequence to determine the prediction order of each token $z_i$. The Halton sequence, a low-discrepancy sequence, ensures uniform token coverage across the 2D spatial structure of the image while reducing the mutual information shared within each set. Consequently, we retain the bidirectional attention mechanism, which leverages the 2D nature of images to facilitate global contextual understanding, akin to Transformer encoder-style models. Finally, we adopt an arccos scheduling scheme to progressively increase the number of tokens within each group $\mathcal{Z}_s$, thereby balancing uniform token distribution with efficient sampling dynamics.

## 4 EXPERIMENTS

### 4.1 IMPLEMENTATION DETAILS

**Model Architecture.** Our models are based on the same repository as (Besnier et al., 2025) with minimal changes for tokenizer, modality and loss adaptation. We use AdaLN for class conditioning similar to the DiT Peebles & Xie (2023). The complete hyper-parameter details are in Appendix F.

**Evaluation Metrics.** To assess image generation quality, we use Fréchet Inception Distance (FID) (Heusel et al., 2017), Inception Score (IS) and the Precision and Recall. For video generation, we rely on Fréchet Video Distance (FVD) (Unterthiner et al., 2019).

**Modalities.** For class-to-image generation on ImageNet, we use a pre-trained LlamaGen tokenizer (Sun et al., 2024), with a downscale spatial factor of 16 and a codebook of 16,384 codes. On Cifar10 (Krizhevsky et al., 2009), we do not use any learnable tokenizer. Instead, images are quantized, mapping each RGB pixel to a single token using the formula: $R + G \cdot q + B \cdot q^2$ with a codebook size of $q^3$. We set $q = 16$, yielding a codebook size of 4,096. For class-to-video generation on UCF-101 (Soomro et al., 2012), we use OmniTokenizer (Wang et al., 2024a) to encode 17-frame videos with a codebook of 8,192 codes. Finally, for NuScenes (Caesar et al., 2020), each frame is tokenized independently using LlamaGen. We use a single frame as conditioning (image-to-video) and generate the following 16 frames auto-regressively, with a maximum context of 8 frames.

### 4.2 TOKEN FIXING

**Random Token Injection $\alpha$.** We analyze the impact of varying our main hyper-parameter $\alpha \in \{0, 0.1, 0.2, 0.3\}$ during training, as presented in Table 1. This parameter plays two key roles in the network's behavior. During training, it controls the amount of randomly injected tokens in the context, influencing the model's ability to handle noisy inputs. When $\alpha > 0$, the model learns to be

| Dataset | **ImageNet** | | **ImageNet** | | **CIFAR-10** | | **UCF-101** | **NuScenes** |
|---|---|---|---|---|---|---|---|---|
| Model | Small | | Base | | Small | | Base | Base |
| Tokenizer | LlamaGen | | LlamaGen | | – | | OmniTok | LlamaGen |
| $\alpha$ | FID50k↓ | IS↑ | FID50k↓ | IS↑ | FID10k↓ | IS↑ | FVD↓ | FVD↓ |
| 0.0 | 46.86 | 23.40 | 25.30 | 48.12 | 26.53 | 10.69 | 558.19 | 529.5 |
| 0.1 | 36.03 | 30.56 | 20.65 | 54.98 | **20.78** | **11.87** | 327.61 | 502.7 |
| 0.2 | **32.47** | 34.87 | **19.83** | 59.49 | 22.26 | 11.62 | **270.15** | **476.9** |
| 0.3 | 33.57 | **35.42** | 21.03 | **63.17** | 23.23 | 11.61 | 316.30 | 515.7 |

Table 1: **Random token injection:** Effect of varying random token injection $\alpha$ after 410k training steps. Across cls-to-img datasets (ImageNet$_{train}$, CIFAR-10$_{val}$), cls-to-video (UCF-101$_{test}$), and img-to-video (NuScenes$_{train+val}$), our framework consistently improves performance when $\alpha > 0$, which enables self-correction during sampling. Best results are highlighted in bold.

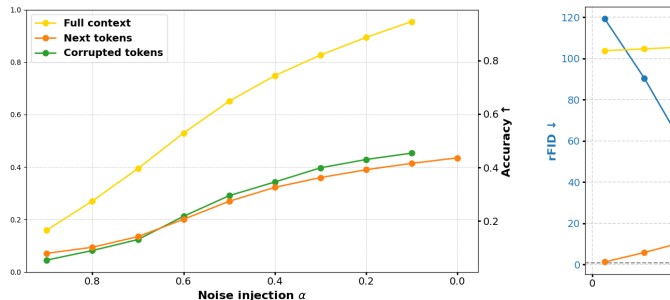 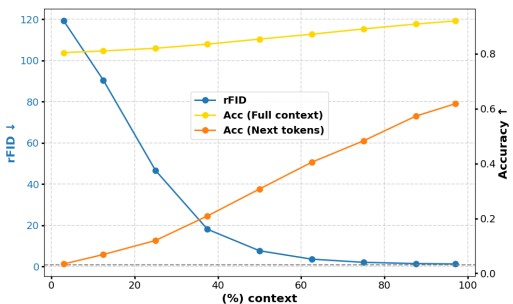

(a) **Prediction Accuracy and rFID using only 37% tokens from a real image in context.** Each available token is then replace by a random tokens based on $\alpha$. Fixing corrupted tokens is as challenging as predicting next tokens.

(b) **Prediction Accuracy and rFID using constant $\alpha = 0.2$.** We track the accuracy of one-step reconstructions as the proportion of the context increase. Most errors occur in the early stages, when only a few tokens are available in the context.

Figure 3: **One-step reconstruction on ImageNet 256×256 validation set.** Evaluation of model accuracy and fidelity under corrupted token contexts.

robust against perturbations. During sampling, setting $\alpha > 0$ enables the model to detect erroneous samples from previous iterations and refine predictions by better estimating the distribution over the input tokens. Analysis shows that increasing $\alpha$ up to $0.2$ consistently improves all metrics, on all datasets modalities, and resolutions; the benefit plateaus for higher values. This suggests that a controlled level of token injection at training time helps to improve the quality of the samples.

**Model Accuracy.** We evaluate in Figure 3 whether our model can accurately correct corrupted tokens and predict the next tokens among the 16,384 codebook entries. In Figure 3a, we keep only 37% tokens in the context, in which $n \times \alpha$ random tokens are injected. We then perform a single prediction step and measure whether the true label is among the top 1% of predicted probabilities, reporting the metrics as Top-1% accuracy. Specifically, we compute: (i) Acc$_{next}$, for predicting the next tokens; (ii) Acc$_{full\_context}$, for all tokens in the context; and (iii) Acc$_{corrupted\_tokens}$, computed only over the perturbed tokens in the context. Our best model with $\alpha = 0.2$ shows strong performance even under high perturbation. Moreover, it shows that the model handles unchanged tokens well, while fixing corrupted ones is as challenging as predicting next tokens, see discrepancy between green and yellow curve. In Figure 3b, most errors occur early, while later predictions improve, highlighting the need for a correction mechanism.

**Self-Correction.** Here we keep all tokens in the context and demonstrate Figure 4 the model's ability to correct randomly injected tokens in the image. The model successfully recovers tokens in a single step even when $\alpha$ is set higher than the values used during training. We test the model's ability to correct corrupted latent codes by copying a crop of GT tokens from one region and pasting them into a different location within the latent space. The modified latent code is passed through

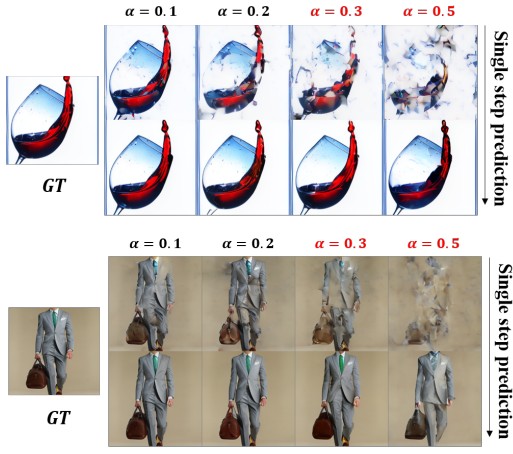

Figure 4: **Random token correction:** Top rows: random token injected at different values of $\alpha$, with no other tokens masked. Bottom rows: the model's prediction in a single step. The figure illustrates the model's ability to accurately correct tokens even when $\alpha$ exceeds the training range (i.e., $\alpha > 0.2$). Better visible when zoomed in.

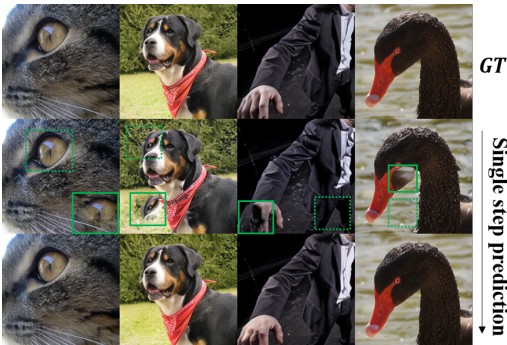

Figure 5: **Patch tokens correction:** Tokens from the green dotted square are copied and pasted into the solid green square, creating a corrupted image (second row). Only the tokens in the solid green square are altered, while all other tokens remain unchanged. BIGFix is able to correct these tokens in a single step (third row), demonstrating its ability to perform correction under out-of-distribution latent manipulations. Better visible when zoomed in.

BIGFix, which fixes the token values in a single step. The model effectively corrects artifacts and local inconsistencies, such as missing fingers or duplicate eyes, as illustrated in Figure 5.

Visualization in Figure 1 showcases the model's capacity to self-correct during sampling from scratch. For instance, it successfully prevents the generation of an extra mouth for the shark. On average, the model corrects 58 tokens per image over 32 sampling steps, corresponding to approximately 10% of the final 576 tokens.

### 4.3 IMAGE SYNTHESIS COMPARISONS

We evaluate across different model sizes to assess their impact on performance, as shown in Table 2. Our model demonstrates faster convergence compared to Diffusion (Peebles & Xie, 2023), Flow Matching (Ma et al., 2024), and auto-regressive approaches (Sun et al., 2024) without the need of representation alignment (Yu et al., 2025b). Using only 16 steps, we outperform them consistently, given comparable training steps and parameter counts.

We compare our method against SOTA in Table 3. We use LlamaGEN to encode images in $24 \times 24$ visual tokens, producing an initial resolution of 384, which we downsampled to 256 following Sun et al. (2024). We train our model with $\alpha = 0.2$, 32 steps, the Halton sequence, and the arccos scheduling. The full ablation is available in see Appendix D. When we prevent our model from correcting during sampling, we achieve an FID of 3.36, 0.38 better than the comparable Halton-MaskGIT baseline. Random token injection acts as data augmentation, enhancing robustness. Self-correction improves the FID by 0.87, reaching 2.49. Our approach is among the fastest sampling methods, as we require only 32 steps. Compared to VAR, we better cover the diversity of the real data (Recall). Additional results for UCF-101 and NuScenes are available in Appendix C.

We present qualitative results in Figure 1, Figure 6, and Appendix J. BIGFix shows strong visual fidelity, effective error correction, and high diversity, highlighting the effectiveness of our approach.

### 5 CONCLUSION

After Theoros built his new image synthesis framework, Eritos asks him if his idea worked.

**T**: "Absolutely! We introduced BIGFix a bidirectional image generation framework with token correction, addressing key limitations of existing multi-token prediction methods. During training,

| Model | #Para. | Training | Steps | FID50k↓ | IS↑ |
|---|---|---|---|---|---|
| DiT-S/2 (Peebles & Xie, 2023) | 33M | 400k | 250 | 68.40 | - |
| SiT-S/2 (Ma et al., 2024) | 33M | 400k | 250 | 57.60 | - |
| Halton-MaskGIT-S (Besnier et al., 2025) | 69M | 410k | 32 | 38.49 | 32.80 |
| **BIGFix-Small (our)** | 50M | 410k | **16** | **31.27** | **37.79** |
| DiT-B/2 (Peebles & Xie, 2023) | 130M | 400k | 250 | 43.47 | - |
| SiT-B/2 (Ma et al., 2024) | 130M | 400k | 250 | 33.50 | - |
| LlamaGen-B (Sun et al., 2024) | 111M | 530k | 256 | 33.44 | 37.53 |
| Halton-MaskGIT-B (Besnier et al., 2025) | 142M | 410k | 32 | 24.91 | 58.98 |
| **BIGFix-Base (our)** | 143M | 410k | **16** | **19.83** | **59.49** |
| DiT-L/2 (Peebles & Xie, 2023) | 458M | 400k | 250 | 23.30 | - |
| SiT-L/2 (Ma et al., 2024) | 458M | 400k | 250 | 18.80 | - |
| LlamaGen-L (Sun et al., 2024) | 343M | 530k | 256 | 19.07 | 64.35 |
| Halton-MaskGIT-L (Besnier et al., 2025) | 480M | 410k | 32 | 14.31 | 84.32 |
| **BIGFix-Large (our)** | 480M | 410k | **16** | **11.36** | **95.17** |
| DiT-XL/2 (Peebles & Xie, 2023) | 675M | 400k | 250 | 19.50 | - |
| SiT-XL/2 (Ma et al., 2024) | 675M | 400k | 250 | 17.20 | - |
| LlamaGen-XL (Sun et al., 2024) | 775M | 530k | 256 | 18.04 | 69.88 |
| **BIGFix-XLarge (our)** | 693M | 410k | **16** | **9.25** | **103.60** |

Table 2: **Scaling law on class-conditional ImageNet 256×256 benchmark.** Analysis of model sizes, without classifier-free guidance. Our method converges faster than other competitive methods.

| Type | Model | #Para. | Step | Training | FID↓ | IS↑ | Prec.↑ | Rec.↑ |
|---|---|---|---|---|---|---|---|---|
| Conti. | LDM-4 (Rombach et al., 2022) | 400M | 250 | 0.2M | 3.60 | 247.7 | — | — |
| | DiT-XL/2 (Peebles & Xie, 2023) | 675M | 250 | 7.0M | 2.27 | **278.2** | **0.83** | 0.57 |
| | SiT-XL/2 (Ma et al., 2024) | 675M | 250 | 7.0M | **2.06** | 270.3 | 0.82 | **0.59** |
| AR | Open-MAGVIT2-L Luo et al. (2024) | 804M | 256 | 2.0M | **2.51** | 271.7 | **0.84** | 0.54 |
| | LlamaGen-XL (Sun et al., 2024) | 775M | 576 | 1.6M | 2.62 | 244.1 | 0.80 | **0.57** |
| NEW→ | ZipAR (He et al., 2025a) | 775M | 331 | - | 3.67 | - | - | - |
| | VAR (Tian et al., 2024) | 600M | 10 | 2.0M | 2.57 | **302.6** | 0.83 | 0.56 |
| MIM | MaskGIT (Chang et al., 2022) | 227M | 12 | 1.6M | 6.18 | 182.1 | 0.80 | 0.51 |
| | TokenCritics (Lezama et al., 2022) | 454M | 36 | 3.2M | 4.69 | 174.5 | 0.76 | 0.53 |
| NEW→ | NAR (He et al., 2025b) | 816M | 31 | - | 2.70 | 277.5 | 0.81 | 0.58 |
| Baseline → | Halton MaskGIT (Besnier et al., 2025) | 705M | 32 | 2.0M | 3.74 | **279.5** | 0.81 | 0.60 |
| Ours | **BIGFix-XLarge** - *No cfg* | 693M | 32 | 2.0M | 6.06 | 145.9 | 0.75 | 0.65 |
| | **BIGFix-XLarge** - *No Correction* | 693M | 32 | 2.0M | 3.36 | 246.3 | 0.80 | 0.61 |
| | **BIGFix-XLarge** | 693M | 32 | 2.0M | **2.49** | 252.5 | **0.83** | **0.63** |

Table 3: **SOTA table on class-conditional ImageNet 256×256 benchmark.** We only include models with sizes below 1B. BIGFix-XLarge improves over MIM methods.

we inject *random tokens* to enhance the model's robustness to errors. The model learns to detect inconsistencies in the context, fix them, and predict the next tokens of the sequence. When sampling, rather than simply generating new tokens at each step, our method actively refines the context, progressively fixing errors introduced in earlier stages. Our method achieves consistent improvements in image synthesis on ImageNet and CIFAR-10, as well as in video generation on UCF-101 and NuScenes. BIGFix method provides a better balance of speed and quality for visual synthesis.

**E**: "Amazing, I guess this means I'll be seeing more chicken images from you soon with this?"

**T**: "You bet. Get ready for the most photorealistic generated chicken ever! (see Figure 12b)"

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

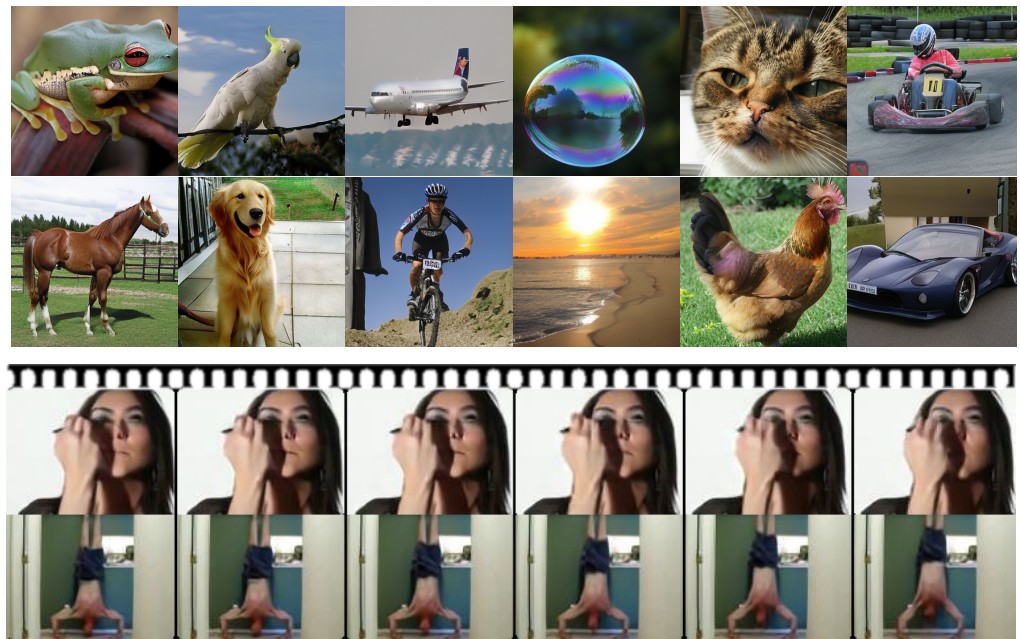

Figure 6: **Qualitative Results:** The first two rows showcase selected samples from our largest model on ImageNet 256×256, while the last two rows feature generated videos from UCF-101. Both modalities demonstrate strong visual fidelity and coherence.

Huiwen Chang, Han Zhang, Jarred Barber, Aaron Maschinot, Jose Lezama, Lu Jiang, Ming-Hsuan Yang, Kevin Patrick Murphy, William T Freeman, Michael Rubinstein, et al. Muse: Text-to-image generation via masked generative transformers. In *ICML*, 2023. 2, 3, 18

Soravit Changpinyo, Piyush Sharma, Nan Ding, and Radu Soricut. Conceptual 12M: Pushing web-scale image-text pre-training to recognize long-tail visual concepts. In *CVPR*, 2021. 15

Junsong Chen, Chongjian Ge, Enze Xie, Yue Wu, Lewei Yao, Xiaozhe Ren, Zhongdao Wang, Ping Luo, Huchuan Lu, and Zhenguo Li. Pixart-$\sigma$: Weak-to-strong training of diffusion transformer for 4k text-to-image generation. In *ECCV*, 2024a. 16

Junsong Chen, Jincheng YU, Chongjian GE, Lewei Yao, Enze Xie, Zhongdao Wang, James Kwok, Ping Luo, Huchuan Lu, and Zhenguo Li. Pixart-$\alpha$: Fast training of diffusion transformer for photorealistic text-to-image synthesis. In *The Twelfth International Conference on Learning Representations*, 2024b. URL https://openreview.net/forum?id=eAKmQPe3m1. 15, 16

Mark Chen, Alec Radford, Rewon Child, Jeffrey Wu, Heewoo Jun, David Luan, and Ilya Sutskever. Generative pretraining from pixels. In *ICML*, 2020. 2, 3

Hyung Won Chung, Le Hou, Shayne Longpre, Barret Zoph, Yi Tay, William Fedus, Yunxuan Li, Xuezhi Wang, Mostafa Dehghani, Siddhartha Brahma, et al. Scaling instruction-finetuned language models. *JMLR*, 2024. 15

Thomas M Cover. *Elements of information theory*. John Wiley & Sons, 1999. 15

Jacob Devlin, Ming-Wei Chang, Kenton Lee, and Kristina Toutanova. Bert: Pre-training of deep bidirectional transformers for language understanding. In *NAACL*, 2019. 3

Prafulla Dhariwal and Alexander Nichol. Diffusion models beat gans on image synthesis. *NeurIPS*, 2021. 3

Nicolas Dufour, Victor Besnier, Vicky Kalogeiton, and David Picard. Don't drop your samples! coherence-aware training benefits conditional diffusion. In *CVPR*, 2024. 15, 16

Nicolas Dufour, Lucas Degeorge, Arijit Ghosh, Vicky Kalogeiton, and David Picard. Miro: Multi-reward conditioned pretraining improves t2i quality and efficiency. *arXiv:2510.25897*, 2025. 15

Patrick Esser, Robin Rombach, and Björn Ommer. Taming transformers for high-resolution image synthesis. In *CVPR*, 2020. 2, 3

Patrick Esser, Sumith Kulal, Andreas Blattmann, Rahim Entezari, Jonas Müller, Harry Saini, Yam Levi, Dominik Lorenz, Axel Sauer, Frederic Boesel, et al. Scaling rectified flow transformers for high-resolution image synthesis. In *ICLM*, 2024. 3, 15, 16

Shenyuan Gao, Jiazhi Yang, Li Chen, Kashyap Chitta, Yihang Qiu, Andreas Geiger, Jun Zhang, and Hongyang Li. Vista: A generalizable driving world model with high fidelity and versatile controllability. In *NeurIPS*, 2024. 18

Ian Goodfellow, Jean Pouget-Abadie, Mehdi Mirza, Bing Xu, David Warde-Farley, Sherjil Ozair, Aaron Courville, and Yoshua Bengio. Generative adversarial networks. *NeurIPS*, 2014. 2, 3

Yefei He, Feng Chen, Yuanyu He, Shaoxuan He, Hong Zhou, Kaipeng Zhang, and Bohan Zhuang. Zipar: Accelerating auto-regressive image generation through spatial locality. *ICML*, 2025a. 4, 9

Yefei He, Yuanyu He, Shaoxuan He, Feng Chen, Hong Zhou, Kaipeng Zhang, and Bohan Zhuang. Neighboring autoregressive modeling for efficient visual generation. In *ICCV*, 2025b. 4, 9

Martin Heusel, Hubert Ramsauer, Thomas Unterthiner, Bernhard Nessler, and Sepp Hochreiter. Gans trained by a two time-scale update rule converge to a local nash equilibrium. In *NeurIPS*, 2017. 6

Jonathan Ho, Ajay Jain, and Pieter Abbeel. Denoising diffusion probabilistic models. In *NeurIPS*, 2020. 2

Jonathan Ho, Tim Salimans, Alexey Gritsenko, William Chan, Mohammad Norouzi, and David J Fleet. Video diffusion models. *NeurIPS*, 2022. 3

Jordan Hoffmann, Sebastian Borgeaud, Arthur Mensch, Elena Buchatskaya, Trevor Cai, Eliza Rutherford, Diego de Las Casas, Lisa Anne Hendricks, Johannes Welbl, Aidan Clark, et al. Training compute-optimal large language models. In *NeurIPS*, 2022. 2

Yang Jin, Zhicheng Sun, Ningyuan Li, Kun Xu, Kun Xu, Hao Jiang, Nan Zhuang, Quzhe Huang, Yang Song, Yadong Mu, and Zhouchen Lin. Pyramidal flow matching for efficient video generative modeling. In *ICLR*, 2024. 3

Minguk Kang, Jun-Yan Zhu, Richard Zhang, Jaesik Park, Eli Shechtman, Sylvain Paris, and Taesung Park. Scaling up gans for text-to-image synthesis. In *CVPR*, 2023. 3

Diederik P. Kingma and Max Welling. Auto-encoding variational bayes. In *ICLR*, 2014. 2

Yuval Kirstain, Adam Polyak, Uriel Singer, Shahbuland Matiana, Joe Penna, and Omer Levy. Pick-a-pic: An open dataset of user preferences for text-to-image generation. *NeurIPS*, 2023. 15

Alex Krizhevsky, Geoffrey Hinton, et al. Learning multiple layers of features from tiny images, 2009. 6

Black Forest Labs. Flux, 2024. 2, 15, 16

Daesoo Lee, Erlend Aune, and Sara Malacarne. Masked generative modeling with enhanced sampling scheme. *arXiv: 2309.07945*, 2023. 4

José Lezama, Huiwen Chang, Lu Jiang, and Irfan Essa. Improved masked image generation with token-critic. In *ECCV*, 2022. 2, 4, 9

Yaron Lipman, Ricky T. Q. Chen, Heli Ben-Hamu, Maximilian Nickel, and Matthew Le. Flow matching for generative modeling. In *ICLR*, 2023. 2, 3

Haotian Liu, Chunyuan Li, Yuheng Li, Bo Li, Yuanhan Zhang, Sheng Shen, and Yong Jae Lee. Llavanext: Improved reasoning, ocr, and world knowledge, 2024. 15

Haozhe Liu, Bing Li, Haoqian Wu, Hanbang Liang, Yawen Huang, Yuexiang Li, Bernard Ghanem, and Yefeng Zheng. Combating mode collapse via offline manifold entropy estimation. In *AAAI*, 2023. 3

Jiachen Lu, Ze Huang, Zeyu Yang, Jiahui Zhang, and Li Zhang. Wovogen: World volume-aware diffusion for controllable multi-camera driving scene generation. In *ECCV*, 2024. 18

Zhuoyan Luo, Fengyuan Shi, Yixiao Ge, Yujiu Yang, Limin Wang, and Ying Shan. Open-magvit2: An open-source project toward democratizing auto-regressive visual generation. *arXiv preprint arXiv:2409.04410*, 2024. 9

Nanye Ma, Mark Goldstein, Michael S Albergo, Nicholas M Boffi, Eric Vanden-Eijnden, and Saining Xie. Sit: Exploring flow and diffusion-based generative models with scalable interpolant transformers. In *ECCV*, 2024. 8, 9

Fabian Mentzer, David Minnen, Eirikur Agustsson, and Michael Tschannen. Finite scalar quantization: Vq-vae made simple. In *ICLR*, 2024. 2

William Peebles and Saining Xie. Scalable diffusion models with transformers. In *ICCV*, 2023. 6, 8, 9

Dustin Podell, Zion English, Kyle Lacey, Andreas Blattmann, Tim Dockhorn, Jonas Müller, Joe Penna, and Robin Rombach. SDXL: Improving latent diffusion models for high-resolution image synthesis. In *ICLR*, 2024. 16

Alec Radford, Jeffrey Wu, Rewon Child, David Luan, Dario Amodei, Ilya Sutskever, et al. Language models are unsupervised multitask learners. *OpenAI blog*, 2019. 2, 3

Alec Radford, Jong Wook Kim, Chris Hallacy, Aditya Ramesh, Gabriel Goh, Sandhini Agarwal, Girish Sastry, Amanda Askell, Pamela Mishkin, Jack Clark, et al. Learning transferable visual models from natural language supervision. In *ICML*, 2021. 15

Aditya Ramesh, Mikhail Pavlov, Gabriel Goh, Scott Gray, Chelsea Voss, Alec Radford, Mark Chen, and Ilya Sutskever. Zero-shot text-to-image generation. In *ICML*, 2021. 3

Aditya Ramesh, Prafulla Dhariwal, Alex Nichol, Casey Chu, and Mark Chen. Hierarchical text-conditional image generation with clip latents. *arXiv:2204.06125*, 2022. 3

Robin Rombach, Andreas Blattmann, Dominik Lorenz, Patrick Esser, and Bjorn Ommer. High-resolution image synthesis with latent diffusion models. In *CVPR*, 2022. 3, 9, 16, 18

Tim Salimans and Jonathan Ho. Progressive distillation for fast sampling of diffusion models. In *ICLR*, 2022. 4

Christoph Schuhmann, Richard Vencu, Romain Beaumont, Robert Kaczmarczyk, Clayton Mullis, Aarush Katta, Theo Coombes, Jenia Jitsev, and Aran Komatsuzaki. Laion-400m: Open dataset of clip-filtered 400 million image-text pairs. *arXiv:2111.02114*, 2021. 15

Uriel Singer, Adam Polyak, Thomas Hayes, Xi Yin, Jie An, Songyang Zhang, Qiyuan Hu, Harry Yang, Oron Ashual, Oran Gafni, Devi Parikh, Sonal Gupta, and Yaniv Taigman. Make-a-video: Text-to-video generation without text-video data. In *ICLR*, 2022. 3

Jiaming Song, Chenlin Meng, and Stefano Ermon. Denoising diffusion implicit models. In *ICLR*, 2021. 3

Khurram Soomro, Amir Roshan Zamir, and Mubarak Shah. UCF101: A dataset of 101 human actions classes from videos in the wild. *arXiv 1212.0402*, 2012. 6, 16

Keqiang Sun, Junting Pan, Yuying Ge, Hao Li, Haodong Duan, Xiaoshi Wu, Renrui Zhang, Aojun Zhou, Zipeng Qin, Yi Wang, et al. Journeydb: A benchmark for generative image understanding. *NeurIPS*, 2023. 15

Peize Sun, Yi Jiang, Shoufa Chen, Shilong Zhang, Bingyue Peng, Ping Luo, and Zehuan Yuan. Autoregressive model beats diffusion: Llama for scalable image generation. *arXiv:2406.06525*, 2024. 2, 3, 6, 8, 9

Keyu Tian, Yi Jiang, Zehuan Yuan, Bingyue Peng, and Liwei Wang. Visual autoregressive modeling: Scalable image generation via next-scale prediction. In *NeurIPS*, 2024. 2, 4, 9

Thomas Unterthiner, Sjoerd van Steenkiste, Karol Kurach, Raphaël Marinier, Marcin Michalski, and Sylvain Gelly. Fvd: A new metric for video generation. In *ICLR*, 2019. 6

Aaron Van Den Oord, Oriol Vinyals, et al. Neural discrete representation learning. In *NeurIPS*, 2017. 3

Ruben Villegas, Mohammad Babaeizadeh, Pieter-Jan Kindermans, Hernan Moraldo, Han Zhang, Mohammad Taghi Saffar, Santiago Castro, Julius Kunze, and D. Erhan. Phenaki: Variable length video generation from open domain textual description. In *ICLR*, 2022. 4

Bram Wallace, Meihua Dang, Rafael Rafailov, Linqi Zhou, Aaron Lou, Senthil Purushwalkam, Stefano Ermon, Caiming Xiong, Shafiq Joty, and Nikhil Naik. Diffusion model alignment using direct preference optimization. In *CVPR*, 2024. 4

Hanyu Wang, Saksham Suri, Yixuan Ren, Hao Chen, and Abhinav Shrivastava. LARP: Tokenizing videos with a learned autoregressive generative prior. In *ICLR*, 2025. 16

Junke Wang, Yi Jiang, Zehuan Yuan, Bingyue Peng, Zuxuan Wu, and Yu-Gang Jiang. Omnitokenizer: A joint image-video tokenizer for visual generation. In *NeurIPS*, 2024a. 6, 16

Xiaofeng Wang, Zheng Zhu, Guan Huang, Xinze Chen, Jiagang Zhu, and Jiwen Lu. Drivedreamer: Towards real-world-drive world models for autonomous driving. In *ECCV*, 2024b. 18

Yuqi Wang, Jiawei He, Lue Fan, Hongxin Li, Yuntao Chen, and Zhaoxiang Zhang. Driving into the future: Multiview visual forecasting and planning with world model for autonomous driving. In *CVPR*, 2024c. 18

Xiaoshi Wu, Yiming Hao, Keqiang Sun, Yixiong Chen, Feng Zhu, Rui Zhao, and Hongsheng Li. Human preference score v2: A solid benchmark for evaluating human preferences of text-to-image synthesis. *arXiv:2306.09341*, 2023. 15

Enze Xie, Junsong Chen, Junyu Chen, Han Cai, Haotian Tang, Yujun Lin, Zhekai Zhang, Muyang Li, Ligeng Zhu, Yao Lu, et al. Sana: Efficient high-resolution image synthesis with linear diffusion transformers. *arXiv:2410.10629*, 2024. 16

Jiazheng Xu, Xiao Liu, Yuchen Wu, Yuxuan Tong, Qinkai Li, Ming Ding, Jie Tang, and Yuxiao Dong. Imagereward: Learning and evaluating human preferences for text-to-image generation. *NeurIPS*, 2023. 15

Wilson Yan, Yunzhi Zhang, Pieter Abbeel, and Aravind Srinivas. Videogpt: Video generation using vq-vae and transformers. *arXiv: 2104.10157*, 2021. 3

Jiahui Yu, Xin Li, Jing Yu Koh, Han Zhang, Ruoming Pang, James Qin, Alexander Ku, Yuanzhong Xu, Jason Baldridge, and Yonghui Wu. Vector-quantized image modeling with improved VQ-GAN. In *ICLR*, 2022a. 3

Jiahui Yu, Yuanzhong Xu, Jing Yu Koh, Thang Luong, Gunjan Baid, Zirui Wang, Vijay Vasudevan, Alexander Ku, Yinfei Yang, Burcu Karagol Ayan, Ben Hutchinson, Wei Han, Zarana Parekh, Xin Li, Han Zhang, Jason Baldridge, and Yonghui Wu. Scaling autoregressive models for content-rich text-to-image generation. *arXiv:2206.10789*, 2022b. 3

Jiahui Yu, Yuanzhong Xu, Jing Yu Koh, Thang Luong, Gunjan Baid, Zirui Wang, Vijay Vasudevan, Alexander Ku, Yinfei Yang, Burcu Karagol Ayan, Ben Hutchinson, Wei Han, Zarana Parekh, Xin Li, Han Zhang, Jason Baldridge, and Yonghui Wu. Scaling autoregressive models for content-rich text-to-image generation. *TMLR*, 2022c. 15

Lijun Yu, Yong Cheng, Kihyuk Sohn, José Lezama, Han Zhang, Huiwen Chang, Alexander G. Hauptmann, Ming-Hsuan Yang, Yuan Hao, Irfan Essa, and Lu Jiang. Magvit: Masked generative video transformer. In *CVPR*, 2023. 4, 16

Lijun Yu, Jose Lezama, Nitesh Bharadwaj Gundavarapu, Luca Versari, Kihyuk Sohn, David Minnen, Yong Cheng, Agrim Gupta, Xiuye Gu, Alexander G Hauptmann, Boqing Gong, Ming-Hsuan Yang, Irfan Essa, David A Ross, and Lu Jiang. Language model beats diffusion - tokenizer is key to visual generation. In *ICLR*, 2024. 2, 4, 16

Sihyun Yu, Sangkyung Kwak, Huiwon Jang, Jongheon Jeong, Jonathan Huang, Jinwoo Shin, and Saining Xie. Representation alignment for generation: Training diffusion transformers is easier than you think. In *ICLR*, 2025a. 4

Sihyun Yu, Sangkyung Kwak, Huiwon Jang, Jongheon Jeong, Jonathan Huang, Jinwoo Shin, and Saining Xie. Representation Alignment for Generation: Training Diffusion Transformers Is Easier Than You Think. In *ICLR*, 2025b. 8

Jinjin Zhang, Qiuyu Huang, Junjie Liu, Xiefan Guo, and Di Huang. Diffusion-4k: Ultra-high-resolution image synthesis with latent diffusion models. In *CVPR*, 2025. 15

Wenzhao Zheng, Ruiqi Song, Xianda Guo, Chenming Zhang, and Long Chen. Genad: Generative end-to-end autonomous driving. In *ECCV*, 2024. 18

# A    On Mutual Information minimization

Let the visual latent be $\mathcal{Z} = \{z_1, z_2, z_3, z_4\}$. We would like to model the conditional distribution $P(z_3, z_4 \mid z_1, z_2)$. Because learning the full joint conditional $P(z_3, z_4 \mid z_1, z_2)$ in one step can be difficult, a common approach is to learn the two separate conditionals $P(z_3 \mid z_1, z_2)$ and $P(z_4 \mid z_1, z_2)$ and approximate $P(z_3, z_4 \mid z_1, z_2) \approx P(z_3 \mid z_1, z_2) P(z_4 \mid z_1, z_2)$.

The approximation error introduced by this factorization is composed of the mutual information (Cover, 1999) (MI) between $z_3$ and $z_4$ given the context $(z_1, z_2)$ plus a residual error $E$ due to the underfitting of the model or the (aleatoric) uncertainty in the data. In this paper, we only tackle MI which measures how much knowing $z_3$ reduces our uncertainty about $z_4$, and leave the error $E$ as a training optimization problem. To correct MI, we introduce a noising step and a denoising/fixing step (in parallel with the next tokens prediction, without extra cost).

We define a forward noising function $\Phi$, applied independently to each token:

$$\Phi(\mathcal{Z}) = \mathcal{Z}' = \{z_1', z_2', z_3', z_4'\}, \qquad z_i' = \Phi(z_i; \varepsilon_i),$$

where $\varepsilon_i$ are independent noise sources (This corresponds to Eq. (5)). We then train a model to invert/fix the noisy token by learning the clean distribution using a cross-entropy objective.

$$P(\mathcal{Z} \mid \mathcal{Z}') = P(z_1, z_2, z_3, z_4 \mid z_1', z_2', z_3', z_4'),$$

(This corresponds to Eq. (6).)

In practice, any $\Phi$ that strongly increases MI of the tokens would push the model to learn to decrease the MI (i.e., to fix errors due to token correlations). We use the mapping in Eq. (5), which swaps random tokens within the same image, ensuring high MI between them since tokens from the same image are indeed correlated.

Thus the factorized approximation $P(z_3 \mid z_1', z_2') P(z_4 \mid z_1', z_2')$ plus the fixing step yields a better approximation to the true conditional joint distribution $P(z_3, z_4 \mid z_1, z_2)$.

# B    Text to Image Synthesis

We train BIGFix for text-to-image generation. Our model uses T5-XL (Chung et al., 2024) with standard cross-attention for text conditioning. Following the approach of CAD (Dufour et al., 2024; 2025), we incorporate a coherence score, computed with CLIP (Radford et al., 2021) and Aesthetics 2.5 (), into AdaLN layers to provide additional control over the generation process.

As no common dataset exists in the literature, we train on a combination of CC12M (Changpinyo et al., 2021) and synthetic images from the MidjourneyDB dataset (Sun et al., 2023), resulting in approximately 7M images. The model is initially trained for 500k steps at a resolution of 256×256 with a batch size of 1024. Subsequently, we increase the resolution to 384×384 and train for an additional 50k steps on a combined Laion Aesthetics (Schuhmann et al., 2021) and Aesthetic-4K dataset (Zhang et al., 2025) with a batch size of 256. For Laion and CC12M, 50% of the original captions are replaced with LLaVA-generated captions (Liu et al., 2024), while captions for MidjourneyDB and the 4k dataset are 100% synthetic.

We evaluate our model on the PartiPrompts benchmark (Yu et al., 2022c) using ImageReward (Xu et al., 2023), HPSv2 (Wu et al., 2023), Aesthetics (Schuhmann et al., 2021), and PickAScore (Kirstain et al., 2023).

For ablation studies, we limit the model to BIGFix-Large at 256×256 resolution and batch size of 256. We evaluate performance across multiple values of the parameter $\alpha$ (see Table 4). We find that increasing $\alpha$ consistently improves the visual quality of generated samples. Despite using only 480M parameters, 32 sampling steps, and a relatively modest training set, our model achieves strong performance.

For BIGFix-XLarge at 384×384 resolution, we compare against state-of-the-art methods such as Pixart (Chen et al., 2024b), SD3 (Esser et al., 2024), and Flux (Labs, 2024). Despite being trained on fewer than 8M images with a relatively small model and only 32 sampling steps, our method achieves the best performance for model below 1B parameters, see Table 5.

In Figure 7 and Figure 8, we present selected samples generated by our model on the PartiPrompts benchmark and custom prompt. Across diverse captions, our images demonstrate high fidelity, strong adherence to text prompts, and visually appealing composition.

| Methods | #Para. | Resolution | Train Steps | Step | Aesthetic | ImageReward | HPSv2 | PickScore |
|---|---|---|---|---|---|---|---|---|
| BIGFix ($\alpha$=0.0) | 437M | 256 | 500k | 32 | 5.34 | 0.17 | 0.23 | 0.210 |
| BIGFix ($\alpha$=0.1) | 437M | 256 | 500k | 32 | 5.60 | 0.39 | 0.25 | 0.215 |
| BIGFix ($\alpha$=0.2) | 437M | 256 | 500k | 32 | 5.71 | 0.53 | 0.25 | 0.217 |
| BIGFix ($\alpha$=0.2) | 627M | 384 | 550k | 32 | 5.71 | 0.53 | 0.25 | 0.217 |

Table 4: **Effect of $\alpha$ parameters on text-to-image model.** $\alpha$ consistently improve the quality of the generated sample on `PartiPrompt` benchmark.

| Methods | #Para. | Avg Rank↓ | Aesthetic↑ | ImageReward↑ | HPSv2↑ | PickScore↑ |
|---|---|---|---|---|---|---|
| SD v1.5 (Rombach et al., 2022) | 0.9B | 9.62 | 5.68 | 0.24 | 0.25 | 0.213 |
| SD v2.1 (Rombach et al., 2022) | 0.9B | 8.12 | 5.81 | 0.38 | 0.26 | 0.215 |
| PixArt-$\alpha$ (Chen et al., 2024b) | 0.6B | 4.00 | 6.47 | 0.97 | 0.29 | 0.226 |
| PixArt-$\Sigma$ (Chen et al., 2024a) | 0.6B | 4.37 | 6.44 | 1.02 | 0.29 | 0.225 |
| CAD (Dufour et al., 2024) | 0.4B | 8.37 | 5.56 | 0.69 | 0.26 | 0.214 |
| Sana-1.6B (Xie et al., 2024) | 1.6B | 2.25 | 6.36 | **1.23** | **0.30** | 0.228 |
| SDXL (Podell et al., 2024) | 2.6B | 7.87 | 5.94 | 0.46 | 0.25 | 0.220 |
| SD3-medium (Esser et al., 2024) | 2.0B | 3.87 | 6.18 | 1.15 | 0.30 | 0.225 |
| FLUX-dev (Labs, 2024) | 12B | **1.50** | **6.56** | 1.19 | 0.30 | **0.229** |
| **BIGFix ($\alpha$=0.2) (ours)** | 0.6B | 5.00 | 6.21 | 1.04 | 0.29 | 0.223 |

Table 5: SOTA results on the `PartiPrompt` benchmark. With only 32 sampling steps and fewer than 8M training images, our model achieves the best performance among models with under 1B parameters.

## C VIDEO SYNTHESIS

To test our method beyond image synthesis, we explore class-to-video on UCF101 (Soomro et al., 2012) dataset and img-to-video on NuScenes (Caesar et al., 2020). As demonstrated previously in Table 1, introducing self-correction substantially improves the quality of generated video samples, mirroring the results of our image synthesis experiments.

In Table 6, we compare BIGFix-Large against state-of-the-art video generation models on UCF101. Despite using a smaller model (480M parameters) and fewer training steps (32), our approach achieves a competitive FVD of 242.16. Performance is limited by our reliance on the open-weight OmniTokenizer, which yields a higher rFVD (42) compared to closed-source tokenizers used by MAGVIT (rFVD 25) and MAGVIT2 (rFVD 8.62). This highlights that while our framework is efficient, generative quality remains constrained by tokenization quality and scale. These results demonstrate the potential of BIGFix but indicate that further improvements would require larger models or more advanced tokenizers.

| Model | #Para. | Train (steps) | Steps | FVD↓ |
|---|---|---|---|---|
| MAGVIT Yu et al. (2023)† | 306M | - | 12 | 76 |
| MAGVIT2 (Yu et al., 2024)† | 307M | - | 24 | 58 |
| LARP (Wang et al., 2025)† | 632M | - | 1024 | **57** |
| OmniTokenizer (Wang et al., 2024a) | 650M | 4M | 1280 | 191.14 |
| OmniTokenizer (Wang et al., 2024a) | 227M | 4M | 1280 | 313.14 |
| **BIGFix-Large (our)** | 480M | 410k | 32 | 242.16 |

Table 6: UCF101 results. † use closed source tokenizer.

In Table 7, we report results on the NuScenes dataset. Despite using a relatively small model (BIGFix-Large, 480M parameters) and limited training time (15 hours), our method achieves a competitive FVD of 290.5. Compared to much larger models, such as GenAD (2.7B parameters, FVD

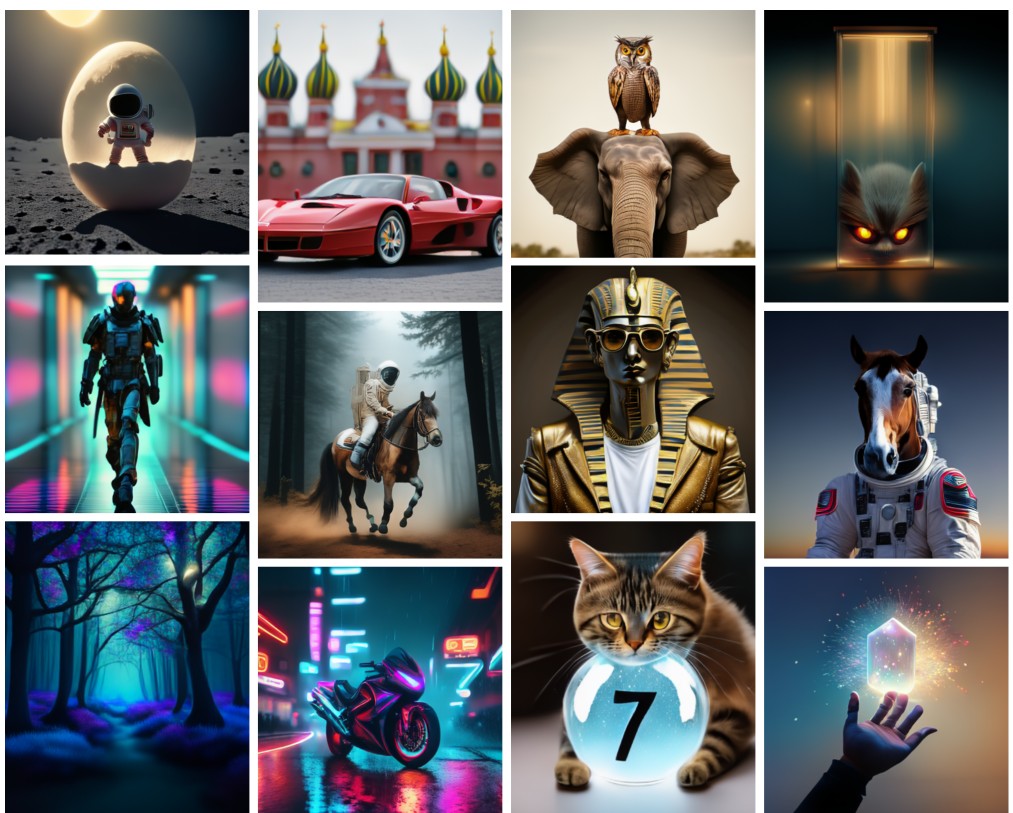

Figure 7: **Selected samples from BIGFix-Xlarge.** With just 32 sampling steps and under 8M training images, the model produces high-quality $384{\times}384$ images.

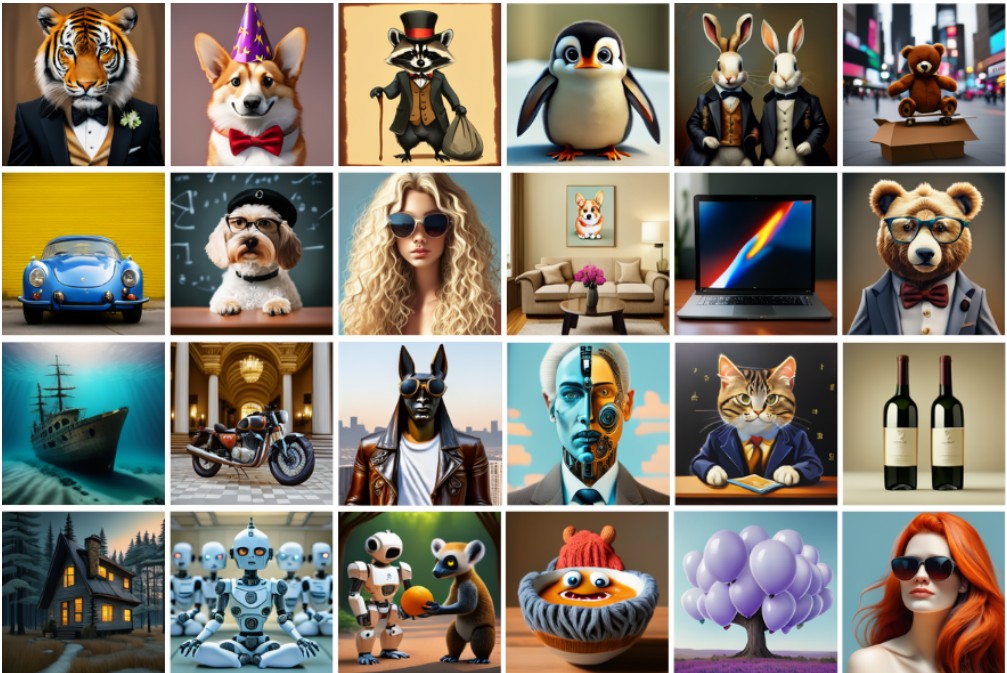

Figure 8: **Randomly generated samples from `PartiPrompt` captions.** BIGFix-Xlarge is capable of producing a wide variety of images, capturing diverse concepts and visual styles from the prompts.

184.0) or Vista (2.5B parameters, FVD 89.4), BIGFix demonstrates strong efficiency and highlights the potential for scaling to larger models and longer training to achieve a lower FVD.

| Model | #Para. | Train (h) | Steps | FVD |
|---|---|---|---|---|
| GenAD (Zheng et al., 2024)‡ | 2.7B | 2 kh | – | 184.0 |
| Vista (Gao et al., 2024)‡ | 2.5B | 1.7 kh | 100 | 89.4 |
| DriveDreamer (Wang et al., 2024b)† | 1.45B | 15 h | – | 452.0 |
| WoVoGen (Lu et al., 2024) | - | 15 h | – | 417.7 |
| Drive-WM (Wang et al., 2024c)† | 1.45B | 15 h | 50 | 122.7 |
| **BIGFix-Large (ours)** | 480M | 15 h | 32+8 | 290.5 |

Table 7: NuScenes results. † use pre-trained weight from SD (Rombach et al., 2022). ‡ Zero-shot FID.

## D   DESIGN CHOICES

In the following section, we investigate random token injection $\alpha$ in Table 8, the number of prediction steps in Table 8, followed by the type of scheduling methods in Table 9, and token ordering in Table 10. Each factor influences efficiency and accuracy, as discussed below.

**Number of Steps**   We investigate in Table 8 the effect of the total number of steps $\{8, 16, 24, 32\}$ to predict the full images. On ImageNet, increasing the number of steps improves performance up to $step = 16$, beyond which the benefits plateau. On the other hand, increasing the number of steps to 24 leads to improved results on Cifar10, suggesting that the step count should be scaled proportionally to the total number of tokens to be predicted. Moreover, increasing $\alpha$ leads the model to correct more tokens during inference, but the number of corrected tokens does not necessarily correlate with FID. Interestingly, the model is more sensitive to $\alpha$ than to the number of decoding steps. For example, a model trained with 8 decoding steps corrects nearly the same number of tokens as a model trained with 32 steps.

**Influence of Scheduling Strategy**   To analyze the effect of different scheduling methods we measure performance across various configurations {square, arccos, linear, root, constant}, and show the results in Table 9. Similarly to Chang et al. (2023) finding, we find out that the arccos scheduling performs the best while concave scheduling performs worse.

**Token Prediction Order**   We compare different token selection strategies {Halton, Spiral, Raster Scan} in Table 10. We find that Halton ordering significantly outperforms raster scan and spiral selection in both metrics. This demonstrates the advantage of structured, but detached, token sampling in guiding the prediction process more effectively. We also compare the performance of 'starting from the same token' location versus 'rolling out the sequence' (Halton+Roll). Specifically, we apply a circular shift to the sequence during both training and testing, enabling the model to begin from any token location in the image. Our results indicate that there is no significant boost in performance between those two strategies.

**Summary of Findings:**   Our ablation study highlights key insights into the impact of different hyper-parameters on model performance Figure 9. We find that using a moderate level of random token injection ($\alpha = 0.2$) drastically improves the performance. Setting the number of prediction steps to between 16 and 32 provides an optimal trade-off between efficiency and quality. Additionally, increasing the number of tokens following an arccos-based scheduling strategy outperforms alternative approaches for guiding token prediction. Finally, leveraging the Halton sequence for the token ordering leads to significantly enhanced image quality and therefore it serves as the baseline method we compare to in the main paper.

| | | ImageNet | | | Cifar10 | |
|---|---|---|---|---|---|---|
| **Step** | $\alpha$ | **FID50k↓** | **IS↑** | **#Avg. Cor NEW↓** | **FID10k↓** | **IS↑** |
| | 0.0 | 42.84 | 29.17 | 0.00 | 86.58 | 7.57 |
| 8 | 0.1 | 38.70 | 33.38 | 4.47 | 80.30 | 7.82 |
| | 0.2 | **34.93** | 36.68 | 15.91 | 67.14 | 8.43 |
| | 0.3 | 34.98 | **37.6** | 38.15 | **66.52** | **8.64** |
| | 0.0 | 43.76 | 26.21 | 0.00 | 39.30 | 9.86 |
| 16 | 0.1 | 35.94 | 32.71 | 7.22 | 26.88 | 11.19 |
| | 0.2 | **31.27** | 37.79 | 22.69 | **25.76** | **11.06** |
| | 0.3 | 32.66 | **39.09** | 50.14 | 27.04 | **11.06** |
| | 0.0 | 45.15 | 24.68 | 0.0 | 30.50 | 10.85 |
| 24 | 0.1 | 35.82 | 31.75 | 8.77 | **20.66** | **11.77** |
| | 0.2 | **31.96** | 35.77 | 26.79 | 22.07 | 11.44 |
| | 0.3 | 31.98 | **37.81** | 58.21 | 22.85 | 11.46 |
| | 0.0 | 46.86 | 23.40 | 0.00 | 26.53 | 10.69 |
| 32 | 0.1 | 36.03 | 30.56 | 10.23 | **20.78** | **11.87** |
| | 0.2 | **32.47** | 34.87 | 30.47 | 22.26 | 11.62 |
| | 0.3 | 33.57 | **35.42** | 65.41 | 23.23 | 11.61 |

Table 8: **ImageNet 256 / Cifar10:** Ablation on the random token injection $\alpha$, the number prediction steps and the number of corrected token during inference, without cfg. We show that enabling token fixing, i.e., $\alpha > 0$, largely improves the metrics, while 16 steps is a good trade-off between FID/IS and compute efficiency.

| Scheduler | FID50k↓ | IS↑ |
|---|---|---|
| root | 39.08 | 32.55 |
| linear | 31.29 | 38.02 |
| cosine | 31.45 | 36.11 |
| square | 31.27 | 37.79 |
| arccos | **29.68** | **40.28** |

Table 9: Ablation on scheduling methods ($\alpha = 0.2$, 16 steps). Convex schedulers like arccos perform best.

| Sequence | FID50k↓ | IS↑ |
|---|---|---|
| Halton | 31.62 | **37.87** |
| Halton + Roll | **31.27** | 37.79 |
| Raster Scan | 43.60 | 34.29 |
| Spiral | 36.34 | 26.71 |

Table 10: Ablation on token ordering on ImageNet 256. Uniform prediction (Halton sequence) improves generation.

# E  SAMPLING PATTERN

An important aspect of our method is the order in which tokens are predicted. In the previous section, we show that the Halton ordering outperforms alternative approaches. Figure 10 illustrates these token sequences on a $16 \times 16$ grid. Notably, the Raster scan method immediately reveals its limitations when applied to a 2D grid, as it enforces a rigid left-to-right, top-to-bottom structure. In contrast, both the Random and Halton sequences achieve a more uniform distribution across the grid, avoiding biases toward specific regions. Compared to random ordering, the Halton sequence is more robust to gaps; for instance, in the random ordering example below, the 17th token is sampled early, while its neighboring tokens are selected much later, leading to a less balanced and structured prediction process.

# F  HYPER-PARAMETERS

In Table 12, we provide the hyper-parameters used for training our class-to-image and class-to-video models on CIFAR-10, ImageNet, UCF101 and NuScenes datasets. The training process differs primarily in the number of steps and batch sizes, reflecting the scale of each dataset. A cosine

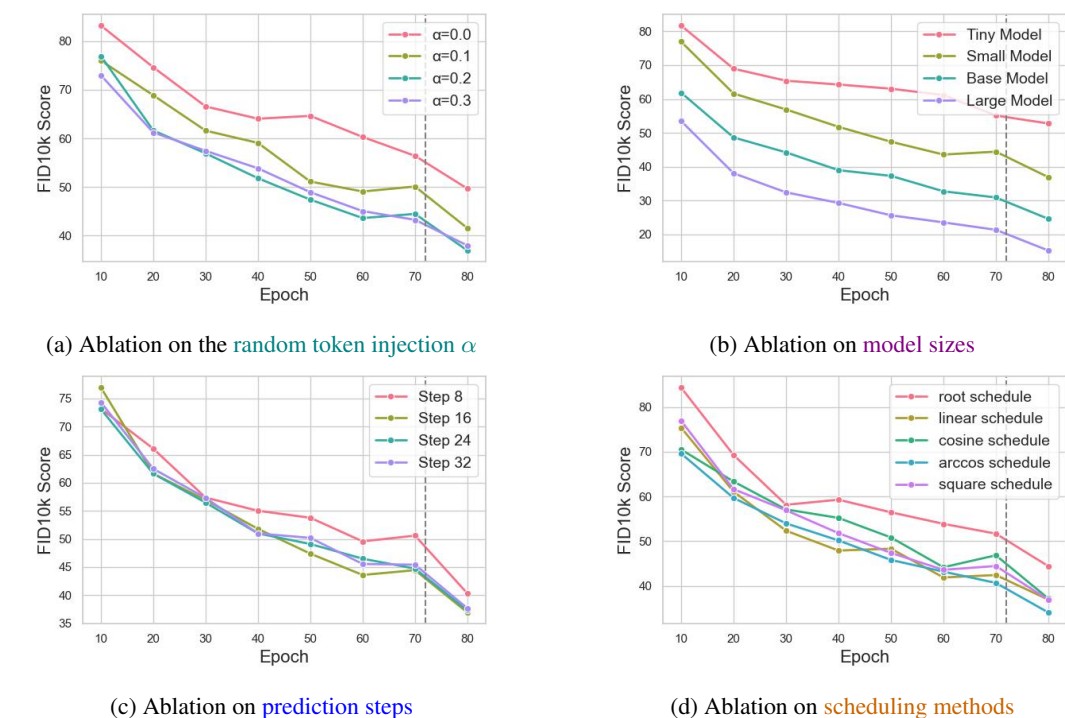

(a) Ablation on the random token injection $\alpha$

(b) Ablation on model sizes

(c) Ablation on prediction steps

(d) Ablation on scheduling methods

Figure 9: **ImageNet 256:** FID10k evolution across model training on ImageNet-256, without cfg and 410k iterations. The moment where learning rate decay was applied is showcased by the dash grey line.

learning rate decay is applied only for the last 10% of the iterations and we use 2,500 warmup steps to stabilize early training. We incorporate gradient clipping (norm = 1) to prevent exploding gradients and classifier-free guidance (CFG) dropout of 0.1 for better sample diversity. The CIFAR-10 model applies horizontal flip augmentation, while no data augmentation is used for ImageNet. Both models are trained using bfloat16 precision for computational efficiency. These hyper-parameters were chosen to ensure stable training while balancing efficiency and performance across different datasets. Finally, we sweep our model size according to Table 11. We also show the training dynamics of the two losses in Figure 11

| Model | Parameters | GFLOPs | Heads | Hidden Dim | Width |
|---|---|---|---|---|---|
| BIGFix-Tiny | 24M | 4.0 | 6 | 384 | 6 |
| BIGFix-Small | 50M | 9.0 | 8 | 512 | 8 |
| BIGFix-Base | 143M | 25.0 | 12 | 768 | 12 |
| BIGFix-Large | 480M | 83.0 | 16 | 1024 | 24 |
| BIGFix-XLarge | 693M | 119.0 | 16 | 1152 | 28 |

Table 11: Transformer model configurations for $16 \times 16$ input size.

## G    INFERENCE SPEED ANALYSIS

We measured inference speed for each model across multiple decoding steps (the mean over 1000 samples with batch size of 8*2 for CFG). From these runs, we report the steps and the throughput (images/sec). The experience is made on a Nvidia A100 and blfoat16 precision and cfg. We do not use KV-cache or any other tricks. The correction does not cause any overhead, but would could prevent KV-caching. We report those values in Table 13

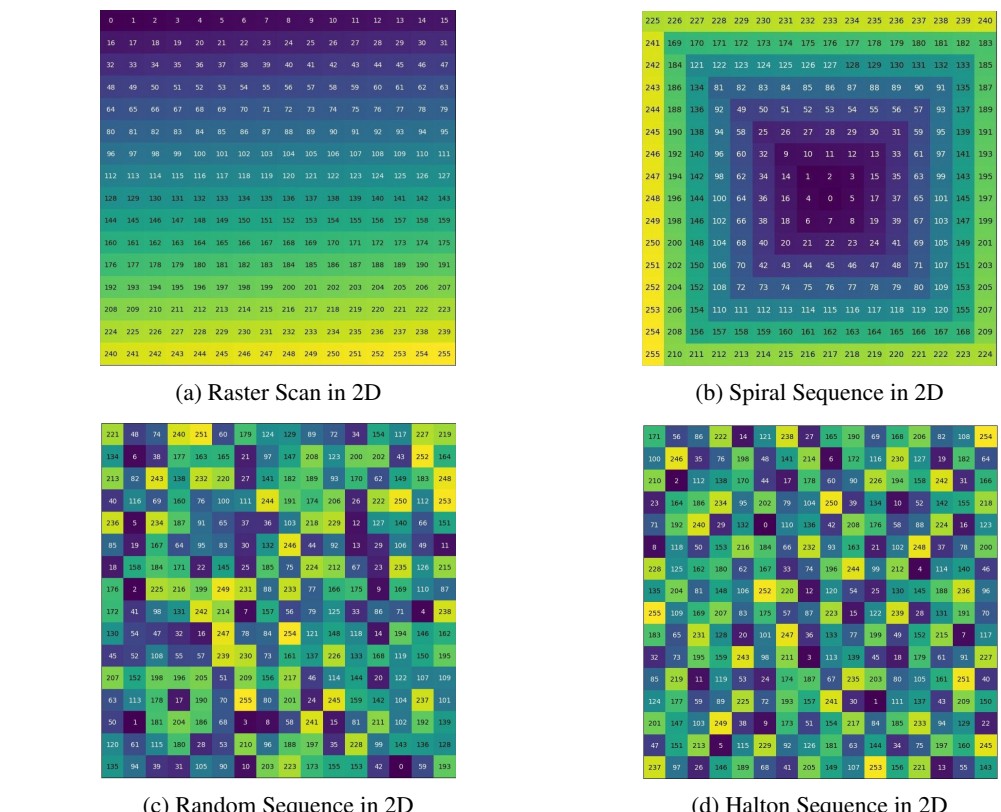

(a) Raster Scan in 2D             (b) Spiral Sequence in 2D

(c) Random Sequence in 2D             (d) Halton Sequence in 2D

Figure 10: Visualization of different sequence orderings in 2D.

| Condition | Cifar10 | ImageNet | UCF101 | NuScenes |
|---|---|---|---|---|
| Training steps | $400k$ | $1.5M$ | $410k$ | $410k$ |
| Batch size | 128 | 256 | 256 | 8 |
| Learning rate | $1 \times 10^{-4}$ | $1 \times 10^{-4}$ | $1 \times 10^{-4}$ | $2 \times 10^{-4}$ |
| Weight decay | 0.03 | 0.03 | 0.03 | 0.03 |
| Optimizer | AdamW | AdamW | AdamW | AdamW |
| Momentum | $\beta_1 = 0.9, \beta_2 = 0.999$ | $\beta_1 = 0.9, \beta_2 = 0.999$ | $\beta_1 = 0.9, \beta_2 = 0.999$ | $\beta_1 = 0.9, \beta_2 = 0.999$ |
| Lr scheduler | Cosine | Cosine | Cosine | Cosine |
| Warmup steps | 2500 | 2500 | 2500 | 2500 |
| Gradient clip norm | 1 | 1 | 1 | 1 |
| CFG dropout | 0.1 | 0.1 | 0.1 | 0.1 |
| dropout | 0.1 | 0.1 | 0.1 | 0.1 |
| Data aug. | Horizontal Flip | No | No | No |
| Precision | bf16 | bf16 | bf16 | bf16 |

Table 12: Hyper-parameters used in the training of class-to-image and class-to-video models.

## H  LIMITATIONS

Like other auto-regressive approaches, BIGFix requires a fixed token unmasking schedule defined at training and maintained at inference, which limits flexibility during prediction. While not an inherent limitation of the method, our experiments were restricted to models below 1 billion parameters to keep computational costs manageable. Extending BIGFix to large-scale image and video generation remains future work.

## I  PROMPTS

For Figure 7, the caption used to generate the images are:

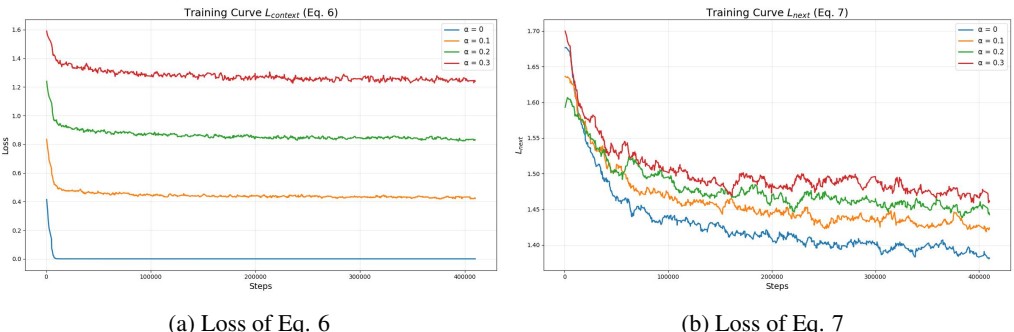

(a) Loss of Eq. 6          (b) Loss of Eq. 7

Figure 11: Training Loss of BIGFix-Small across different values of $\alpha$

| Model | Step | input size $16\times16$ (img/s) | input size $24\times24$ (img/s) |
|---|---|---|---|
| **BIGFix-Tiny** | 8 | 80.63 | 41.98 |
| | 16 | 55.89 | 30.32 |
| | 24 | 42.53 | 23.48 |
| | 32 | 34.45 | 19.30 |
| **BIGFix-Small** | 8 | 71.66 | 37.28 |
| | 16 | 47.18 | 25.51 |
| | 24 | 35.30 | 19.39 |
| | 32 | 28.21 | 15.61 |
| **BIGFix-Base** | 8 | 54.03 | 27.17 |
| | 16 | 33.39 | 16.87 |
| | 24 | 24.16 | 12.21 |
| | 32 | 18.88 | 9.58 |
| **BIGFix-Large** | 8 | 28.18 | 14.37 |
| | 16 | 15.53 | 7.98 |
| | 24 | 10.73 | 5.53 |
| | 32 | 8.21 | 4.22 |
| **BIGFix-XLarge** | 8 | 21.93 | 10.74 |
| | 16 | 11.90 | 5.80 |
| | 24 | 8.15 | 3.97 |
| | 32 | 6.18 | 3.01 |

Table 13: Throughput comparison (img/s) for different input sizes.

- **Image 1:** "A tiny astronaut hatching from a transparent egg on the moon."
- **Image 2:** "A futuristic soldier walking through a bright holographic corridor, colorful reflections on their armor, crisp high-tech atmosphere."
- **Image 3:** "An enchanted forest with bright bioluminescent plants casting blue and purple glows, soft magical lighting throughout the scene."
- **Image 4:** "A Ferrari Testarossa in front of the Kremlin."
- **Image 5:** "A photo of an astronaut riding a horse in the forest."
- **Image 6:** "A futuristic motorcycle speeding through a neon-lit rain-soaked street, bright reflections and vivid colors, dynamic cyberpunk energy."
- **Image 7:** "An owl on top of an elephant's back."
- **Image 8:** "A photograph of a portrait of a statue of a pharaoh wearing steampunk glasses, white t-shirt and leather jacket."
- **Image 9:** "A cat patting a crystal ball with the number 7 written on it in black marker."

- **Image 10:** "A creature barely visible behind glass, soft bright diffused lighting highlighting subtle shapes and glowing eyes, atmospheric."
- **Image 11:** "A horse sitting on an astronaut's shoulders."
- **Image 12:** "A hand reaching toward a floating crystal in softly glowing ambient light, colorful spark-like particles surrounding it, bright fantasy mood."

For Figure 8, the caption used to generate the images are:

- **Image 1:** "A tiger wearing a tuxedo."
- **Image 2:** "A corgi wearing a red bowtie and a purple party hat."
- **Image 3:** "A raccoon wearing formal clothes, wearing a tophat and holding a cane. The raccoon is holding a garbage bag."
- **Image 4:** "A baby penguin."
- **Image 5:** "An oil painting of two rabbits in the style of American Gothic, wearing the same clothes as in the original."
- **Image 6:** "A teddybear on a skateboard in Times Square, doing tricks on a cardboard box ramp."
- **Image 7:** "A blue Porsche 356 parked in front of a yellow brick wall."
- **Image 8:** "A cream-colored labradoodle wearing glasses and black beret teaching calculus at a blackboard."
- **Image 9:** "A girl with long curly blonde hair and sunglasses."
- **Image 10:** "A cozy living room with a painting of a corgi on the wall above a couch and a round coffee table in front of a couch and a vase of flowers on a coffee table."
- **Image 11:** "A laptop."
- **Image 12:** "A nerdy bear wearing glasses and a bowtie."
- **Image 13:** "A sunken ship at the bottom of the ocean."
- **Image 14:** "A motorcycle parked in an ornate bank lobby."
- **Image 15:** "A portrait of a statue of the Egyptian god Anubis wearing aviator goggles, white t-shirt and leather jacket."
- **Image 16:** "Salvador Dalí with a robotic half face."
- **Image 17:** "A super math wizard cat."
- **Image 18:** "Two bottles of wines."
- **Image 19:** "A small house in the wilderness."
- **Image 20:** "Robots meditating."
- **Image 21:** "A robot gives a wombat an orange and a lemur a banana."
- **Image 22:** "A bowl of soup that looks like a monster knitted out of wool."
- **Image 23:** "A tree with leaves that look like purple balloons."
- **Image 24:** "A woman with sunglasses and red hair."

## J ADDITIONAL QUALITATIVE RESULTS

In Figure 12, Figure 13, Figure 14, Figure 15, and Figure 16, we present qualitative results from BIGFix-Large. Without any cherry-picking, but with classifier-free guidance (CFG), we demonstrate that our model is capable of generating realistic and diverse images. Maintaining intricate details on both the objects and the background, using only 24 steps.

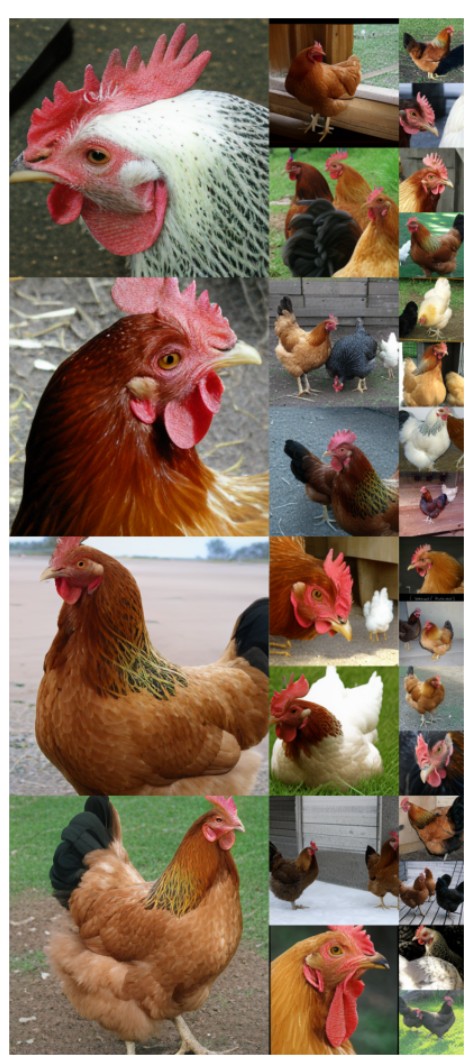

(a) Tree frog **031**                    (b) Chicken **008**

Figure 12: Random samples from our BIGFix-Large model.
$\alpha = 0.2$, $cfg = 4.0$ and 24 steps.

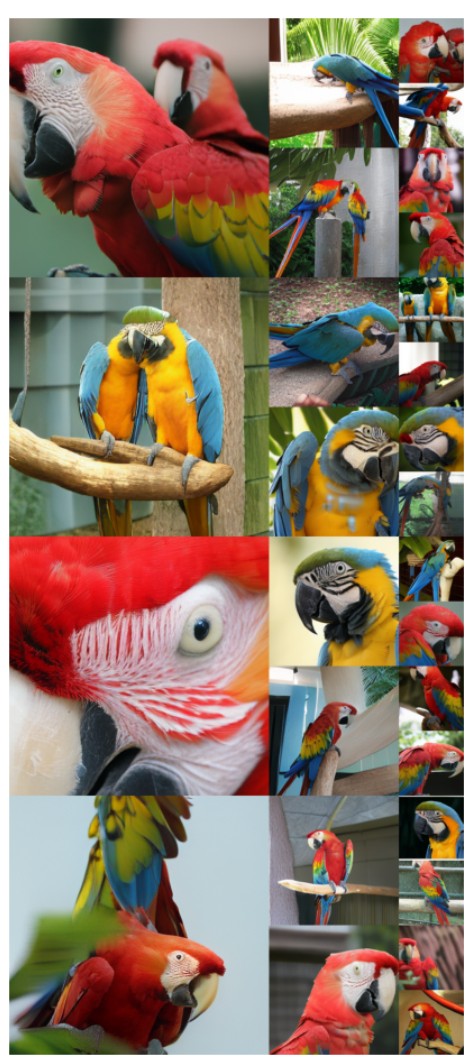

(a) LadyBug **301**          (b) Macaw **88**

Figure 13: Random samples from our BIGFix-Large model.
$\alpha = 0.2$, $cfg = 4.0$ and 24 steps.

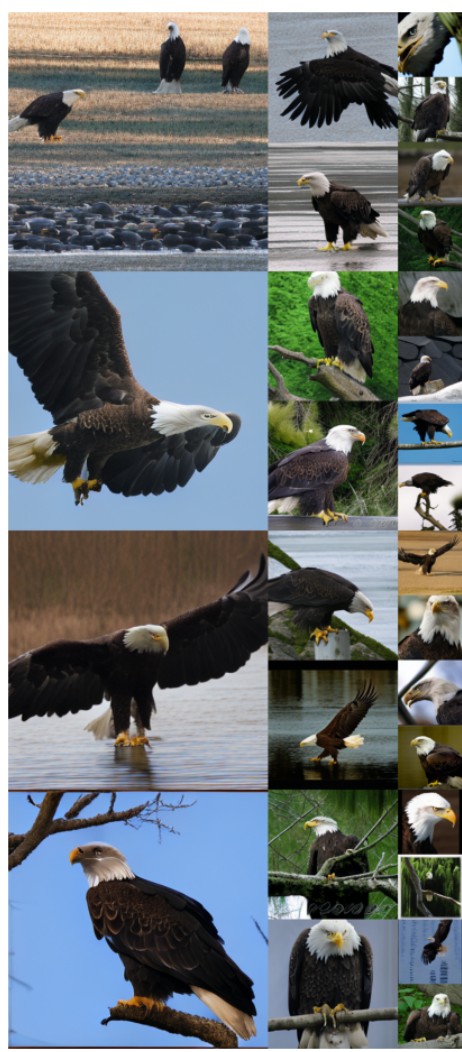

(a) Axolotl **29**                                          (b) Bald Eagle **22**

Figure 14: Random samples from our BIGFix-Large model.
$\alpha = 0.2$, $cfg = 4.0$ and 24 steps.

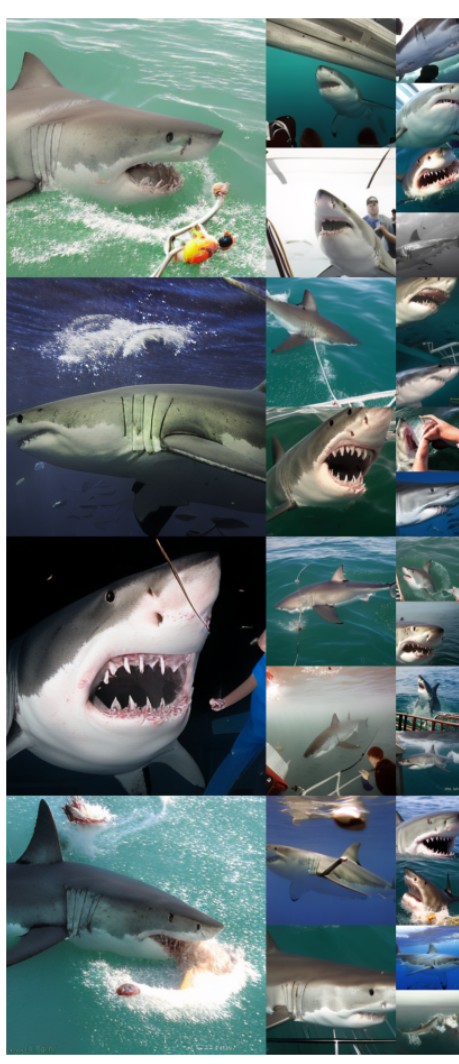

(a) Rock crab **119**                    (b) Great white shark **002**

Figure 15: Random samples from our BIGFix-Large model.
$\alpha = 0.2$, $cfg = 4.0$ and 24 steps.

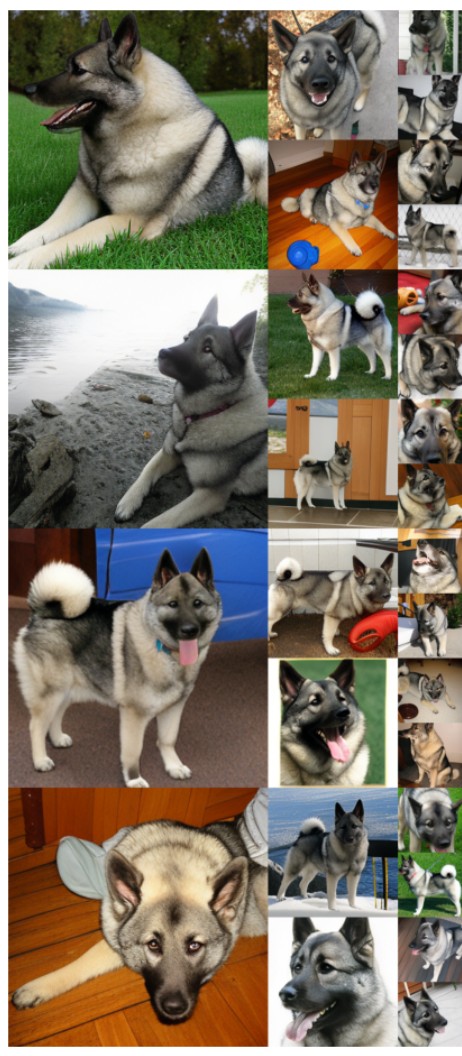

(a) Bobsleigh **450**                              (b) Norwegian Elkhound **174**

Figure 16: Random samples from our BIGFix-Large model.
$\alpha = 0.2$, $cfg = 4.0$ and 24 steps.

