# OpenReview forum: "BIGFIX: BIDIRECTIONAL IMAGE GENERATION WITH TOKEN FIXING"
_ICLR.cc/2026/Conference — Submitted to ICLR 2026_

### Official Review · Reviewer_QMWq · 2025-10-27

**Soundness:** 3
**Presentation:** 2
**Contribution:** 2
**Rating:** 4
**Confidence:** 4

**Summary:**

This submission proposes a fixing mechanism in multi-token prediction AR frameworks for efficient visual generation. The method is first based on a MaskGIT-like AR model with multi-token prediction. During training, some predicted tokens are randomly replaced with random tokens, as a regularization to encourage the model to "fix" these corrupted tokens. By doing so, the proposed method requires fewer inference steps to generate comparable or better image quality. The main results on ImageNet 256x256 generation demonstrates the effectiveness.

**Strengths:**

1. The motivation is clearly stated;

2. The proposed method is generally easy to follow;

3. The writing style of the introduction is novel.

4. The model's ability to correct random tokens is demonstrated clearly.

**Weaknesses:**

1. My biggest concern is that this submission has missed some important references (strong baselines).
    - For example, ZipAR [1] is a training-free multi-token (parallel) AR decoding method. Instead of proposing a model family as this submission, ZipAR can work as a plug-in decoding strategy on various architectures, such as llamagen and Lumina-mGPT. It also reduces the inference steps by >90% with a minor sacrifice of generation quality.
    - NAR [2] further improves the trade-off between efficiency and generation quality.
    - I think these methods should be strong baselines to compare.

2. The advantage and contribution of this work are not stated very clearly. Specifically, if the core advantage is the efficiency (fewer inference steps), the detailed metrics should be reported, such as throughput or latency, GPU memory, etc. Moreover, the trade-off should be compared with the aforementioned baselines.
    - To me, an interesting advantage of this work is its capability to "fix" the corrupted tokens. More analysis could be presented. For example, does the method contribute to better robustness?

3. The evaluated benchmarks are generally small: ImageNet 256x256 class-conditioned generation. Is the method adaptable to text-to-image generation models such as Pixart, SD3 or FLUX? How does it perform when generating higher-resolution images?

4. Some representation might be confusing.
    - For example, where are the blue tokens from in the inference mode in Fig. 2 (bottom)? I understand that they can come from human interference. Without human interference, will the model itself generate erroneous tokens? What is the relationship between the "fixing" capability and the fewer-step inference?
    - The modifications of the model architectures are not clearly stated.

[1] ZipAR: Parallel Autoregressive Image Generation through Spatial Locality.

[2] Neighboring Autoregressive Modeling for Efficient Visual Generation

**Questions:**

Please see the weakness section.

---

> ### Author Response · Authors · 2025-11-21
>
> **Q1: Missing references (ZipAR, NAR)**
>
> Thank you for pointing out these important baselines, we were not aware of them. We have added a dedicated paragraph in the Related Work dedicated to ZipAR [1] and NAR [2]. (See line 174 to 177 in the revised paper and Tab. 3) Table 3.
> ZipAR [2], a method that significantly accelerates inference of AR but it typically causes a small degradation in FID (see Table 1 in ZipAR[1], it increases LlamaGen-XL FID from 2.83 to 3.67). Our method both improves FID while requiring only a small number of forward passes.
> NAR [3], using similar sampling budgets, we obtain better FID (2.58 vs. 2.48), precision (0.82 vs. 0.83), and recall (0.57 vs. 0.55), though IS is lower (293.5 vs. 252.5). Importantly, our model is significantly smaller than the architecture used in NAR.
>
>
> **Q2: Advantage, contribution and robustness:**
>
> Speed is not the intended contribution of this paper. Although our paradigm enables fast parallel multi-token sampling (Appendix D), this is not the conceptual novelty and we do not claim it in the contribution (L100 - L135).
> Our main contribution is summarized in line 100: We inject random tokens during training, teach the model to correct them, and during sampling explicitly allow the model to refine its previous predictions. This mechanism gives the model the ability to ‘fix’ corrupted tokens; the behavior highlighted in the review. Section 4.2 already provides qualitative and quantitative evidence of this correction capability. Our method naturally increases the robustness because it reduces the sensitivity to incorrect tokens appearing during the sampling process, see Figure 4 and 5, where we specifically injected erroneous tokens as well as image patches and observed how our method recovered from it.
>
> **Q3: Small benchmark evaluation, no txt2img, no high resolution**
>
> We conduct experiments on ImageNet 256×256 and 384×384, matching the setups used in LlamaGen. We also evaluate on CIFAR-10 (32×32), which, despite its small image size, has a larger latent spatial resolution (16×16 or 24×24) than the 256×256 ImageNet latents. In addition, we provide results for video synthesis of size 8x288x512 on NuScenes datasets.
> For large-scale text-to-image models (PixArt, SD3, FLUX), reproducing such systems exceeds our compute resources in both dataset scale and training cost. However, in the rebuttal period, we include ablations on CC12M+MidJourneyDB datasets, which offer preliminary insight into the applicability of our approach to text-conditioned generation, while our bigger model is training. See new Table 12 in the revised manuscript.
>
> | Methods            | #Param. | Train Steps | Step | Aesthetic | ImageReward | HPSv2 | PickScore |
> |-------------------|--------|-------------|------|-----------|-------------|-------|-----------|
> | Ours (α=0.0)      | 437M   | 500k        | 32   | 5.34      | 0.17        | 0.23  | 0.210     |
> | Ours (α=0.1)      | 437M   | 500k        | 32   | 5.60      | 0.39        | 0.25  | 0.215     |
> | Ours (α=0.2)      | 437M   | 500k        | 32   | 5.71      | 0.53        | 0.25  | 0.217     |
>
>
>
> **Q4: Clarifying Fig. 2 and the origin of “blue tokens”, relationship between fixing and fewer steps**
> The model does indeed produce erroneous tokens, not due to human intervention, but because it cannot directly model the joint distribution such as P(x_3, x_4 \mid x_1, x_2) in a single step. Instead, it separately learns P(x_3 \mid x_1, x_2) and P(x_4 \mid x_1, x_2), which introduces sampling errors that can be quantified by the mutual information between x_3 and x_4​ conditioned on (x_1, x_2). The refinement step is explicitly designed to correct these errors in subsequent updates. The “blue tokens” in Fig. 2 therefore correspond to model-generated errors—not to human edits. There is no relationship between the capability of fixing the tokens and fewer steps. We do not claim to have fewer steps than the SOTA methods. We keep a similar amount of steps to SOTA and we add the token fixing capability.
> Model architecture modification: We will release the full code upon acceptance to ensure transparency and reproducibility. We reuse the V. Besnier et al. repository, the only architectural modification is the removal of the initial patchifying convolution layer that reduces spatial resolution. All other components remain unchanged.

---

### Official Review · Reviewer_YYtR · 2025-11-01

**Soundness:** 3
**Presentation:** 4
**Contribution:** 3
**Rating:** 8
**Confidence:** 5

**Summary:**

The authors focus on structural inconsistencies that arise in generative vision models, and propose BIGFIX, which adds random tokens from the data distribution in the context and trains the model to correct these, thereby giving the model the ability to fix such errors during denoising. The authors present strong fidelity results on a variety of benchmarks.

**Strengths:**

- The paper is excellently written, with a very clear problem formulation and easy to follow methodology.
- The results on standard image and video benchmarks are particularly strong, making this approach a useful contribution to the community. Fig 5 is a particularly great result, as it shows that even artificially introduced artifacts can be effectively mitigated using this approach.
- The token correction approach, as per my knowledge, is significantly novel in this context.

**Weaknesses:**

- The method uses (Besnier et al., 2025) with Halton scheduling, however do not include it as a baseline in the results. This leads to slightly misleading, potentially stronger impression of the results. Some of the gain that we see in the current Table 2 should also be attributed to the already strong baseline.
- The rate of token injection $\alpha$ is kept fixed during training, the authors mention that a value upto 0.2 leads to improvements. However, would it help more if the alpha value starts with 0 at the beginning of the training process and is scheduled to increase as the training progresses?
- It would be helpful to see the training dynamics of this approach, particularly how are the two losses balanced, what are the scale of these losses, convergence rates, etc. The training curves, if attached to the paper, would be helpful in understanding how token injection modifies the overall training objective and progression empirically.

**Questions:**

Please see the weaknesses section above.

---

> ### Author Response · Authors · 2025-11-21
>
> **Q1: Missing baseline**
>
> In the original submission the method Besnier et al. (2025) (Halton MaskGIT) was present in Table 3 marked as “baseline” (row 443), and we highlighted in bold the improvement obtained by our noise-injection + self-correction mechanism over this baseline, and isolated each component (noise-injection from the self-correction).
> In the rebuttal period, for Table 2, we have run and added three Halton MaskGIT variants trained on ImageNet-256 for 410k steps using their official repository (we keep the HMaskGIT-XL architecture in the main SoTA table):
>
> | Model                              |Trainning Steps | Sampling Steps | FID50k ↓ | IS ↑      |
> |------------------------------------|----------------|----------------|----------|---------  |
> | Halton-MaskGIT-S                   |           410k | 32             | 38.49    | 32.80     |
> | **BIGFix-Small (ours)**                  |  410k          | 16             | **31.27**| **37.79** |
> | Halton-MaskGIT-B                   | 410k           | 32             | 24.91    | 58.98     |
> | **BIGFix-Base (ours)**             | 410k           | 16             | **19.83**| **59.49** |
> | Halton-MaskGIT-L                   | 410k           | 32             | 14.31    | 84.32     |
> | **BIGFix-Large (ours)**            | 410k           | 16             | **11.36**| **95.17** |
>
> These additions ensure that all comparisons are fair and reflect the contribution of our method beyond the strong Halton scheduling baseline. We added these results into Table 2.
>
> **Q2: Scheduling of α during training**
>
> A schedule for α adds extra hyperparameters and complexity but doesn’t yield consistent improvements; in our tests it often performed the same or worse than a fixed α. Since gains already saturate around α=0.2, varying it over training provides no clear benefit. Nevertheless, fine tuning the model on a different target α quickly converges, i.e., one can easily train the model with α=0 and fine-tune for 100k steps to reach the same FID as training from scratch.
>
> **Q3: Training dynamic**
> We did not rebalance the two losses and instead kept a fixed ratio throughout training (i.e., $L= L_{noise} + \lambda * L_{context}$, with $\lambda=1$). We provide the corresponding loss curves in the supplementary material. The code and pretrained models will be released upon acceptance, enabling full reproducibility of the training dynamics and scaling. See new Table 9

---

### Official Review · Reviewer_g51t · 2025-11-02

**Soundness:** 2
**Presentation:** 2
**Contribution:** 1
**Rating:** 4
**Confidence:** 3

**Summary:**

This paper proposes BIGFIX which introduces a token fixing strategy into the Bidirectional Halton ordering Multi-token prediction paradigm for image/video generation tasks. The paper is very easy to follow. However, the reviewer thinks that the method of introducing perturbations into the NTP/MTP paradigm to enhance the robustness of method prediction is very common and not sufficient as the sole core contribution of the paper.

**Strengths:**

The paper is well structured and focuses on one of the most popular topics in autoregression based image generation.

**Weaknesses:**

1. The major weakness is that the contribution and novelty are very limited.
- Introducing noise for improving NTP/MTP paradigm is a commonly used trick in practical scenarios.
- The introduction of the proposed method is as few as the PRELIMINARIES parts (as the Bidirectional Halton ordering is formerly introduced).
- As far as the reviewer's knowledge, SOTA approaches AR-based image/video generation methods show very few structural errors such as supernumerary or missing elements which are illustrated in Figure 1, meaning that the challenges proposed in this paper are somehow out-of-date.
- Besides, the introduced noise still can not resolve the mentioned challenges "On token dependencies." as the error distribution at the inference phase is not exactly the same as the formulated one (sampled from the original image).

2. The introduction section discussion is aimless, the advantage of autoregression based method is not clearly claimed and multiple advanced methods such as next scale prediction for solving the generating order and reducing the generation steps are not discussed.

3. Missing ablation of different image tokenizers. Why are different tokenizers used for different benchmarks?

4. Out-of-date comparison baseline.  All compared baselines are out-of-date as far as the reviewer's knowledge, comparison with the current sota generation model is needed to convince the effectiveness of the proposed method. (such as NAR, Infinity, etc.)

**Questions:**

See weakness parts.

---

> ### Author Response · Authors · 2025-11-21
>
> **Q1: Novelty of token fixing**
>
> - Q1.1 & Q1.2 We would appreciate references supporting this claim, as we are not aware of prior work introducing noise on discrete latent tokens at training time and a fixing mechanism during sampling, specifically within multi-token prediction frameworks for image generation. Existing “noise injection” techniques in diffusion or continuous latent spaces differ fundamentally from our discrete token corruption and correction mechanism.
> - Q1.3 Regarding the structural errors illustrated in Figure 1: these do not appear in purely autoregressive models because they sample tokens sequentially. They arise in parallel or multi-token generation with prior MIM works (e.g., see Fig. 5, 11, 13 in MaskGIT[1], Fig. 12 in Halton MaskGIT[2] or Fig. 5 of Token-Critics[3] where similar artifacts are documented). Our setup explicitly targets this family of models, not classical single token prediction architectures.
> - Q1.4 As noted in our response to Reviewer QMWq, learning to correct erroneous tokens improves the estimation of the joint distribution by allowing the model to refine predictions that cannot be captured in a single parallel sampling step. In other words, we approximate the joint conditional \(P(z_3,z_4 \mid z_1,z_2)\) by factorizing it into separate conditionals, introducing an error due to the mutual information between \(z_3\) and \(z_4\). To reduce this error, we apply a noising function \(\Phi\) and train the model to denoise/fix tokens, improving the factorized approximation to the true joint distribution. Please see Appendix H for more details.
> [1] H Chang: MaskGIT: Masked Generative Image Transformer
> [2] V Besnier: Halton Scheduler for Masked Generative Image Transformer
> [3] J  Lezama: Improved Masked Image Generation with Token-Critic
>
> **Q2: Missing discussion of next-scale prediction.**
>
> To clarify: Our paper does not aim to improve purely autoregressive models. Instead, we focus on multi-token prediction, which is faster but suffers from structural inconsistencies due to parallel sampling. Our contribution is not a new ordering strategy, it is a noise-injection and self-correction mechanism that mitigates the structural defects arising from multi-token prediction. We added a paragraph in the Related Work section for NAR-like methods, but we kindly point out to the reviewer that next-scale is already discussed in lines 171-175 of the original submission.
>
> **Q3: Missing ablations on image tokenizers**
>
> We selected multiple tokenizers to demonstrate that our method is tokenizer-agnostic and can operate across various discrete latent spaces. Conducting a full comparative study of tokenizers is an interesting direction but orthogonal to our contribution. Tokenizers themselves are out of the scope of the paper.
>
> **Q4: Out-of-date baselines**
>
> We will include NAR and parallel AR models in the updated baseline table. Compared to NAR we obtain better FID (2.58 vs. 2.48), precision (0.82 vs. 0.83), and recall (0.57 vs. 0.55), though IS is lower (293.5 vs. 252.5). Importantly, our model is significantly smaller than the architecture used in NAR.
> We kindly ask the reviewer to provide a specific reference for Infinity. To the best of our knowledge, Infinity is a text-to-image architecture, which is not directly comparable to our setting. If the reviewer refers to a different “Infinity” model relevant to discrete image generation, we will be happy to include it once clarified.

---

### Official Review · Reviewer_LeAz · 2025-11-02

**Soundness:** 2
**Presentation:** 2
**Contribution:** 2
**Rating:** 2
**Confidence:** 4

**Summary:**

**Problem.** Multi-token prediction for image generation is fast but brittle: sampling many tokens in parallel introduces incompatible tokens and structural errors that can’t be corrected once committed. **Method.** *BIGFix* trains with **random token injection** into the context and, at inference, **allows backtracking**—the model “fixes” previously sampled tokens while predicting new ones; token order follows a **Halton** low-discrepancy sequence with an **arccos** scheduling of revealed tokens. **Key innovations.** (i) Context corruption via Eq. (5) and dual losses \(L_{\text{context}}+L_{\text{next}}\) (Eqs. 6–7); (ii) bidirectional attention with Halton ordering; (iii) explicit self-correction during sampling. **Main results.** On ImageNet-256, the XL variant reports FID 2.49 in 32 steps versus Halton-MaskGIT 3.74; α≈0.2 improves metrics across ImageNet/CIFAR/UCF-101/NuScenes; A100 latency claim ≈0.25 s/image (2.86× vs LlamaGen-XL). **Significance.** If robust and fairly compared, a “token fixing” mechanism could close quality gaps while retaining the step-efficiency of masked/parallel decoders.

**Strengths:**

- **Clear problem framing** (parallel sampling causes incompatibilities); concrete toy/qualitative cases (Fig. 1, Fig. 5).
- **Simple, general training recipe** (random token injection) with ablations over α and steps (Table 1, Table 4; Fig. 7).
- **Competitive ImageNet results** in few steps; XL gets FID 2.49 @32 steps; speed claims vs AR step counts are plausible (Table 3; Appendix D).

**Weaknesses:**

1) **Correctness of the probabilistic story is underspecified.**
You attribute incompatibilities to “independent sampling” of parallel tokens and approximate \(P(z_a,z_b)\approx P(z_a)P(z_b)\) (Sec. 3.2), but also note the model outputs are “not independent.” Please provide a formal derivation of how the training objective (Eqs. 6–7) induces *error detection* and *targeted correction* under the actual joint decoder, and quantify the *false-positive* rate of overwriting correct tokens during self-correction (see p.xx, Fig. 3).

2) **Self-correction mechanism lacks algorithmic clarity.**
At inference you “allow the model to backtrack,” but the rules are unclear: which tokens are eligible for overwrite, by what criterion (confidence, disagreement, local consistency), and with what schedule? Please add pseudocode describing selection, re-masking, and stopping conditions; report average **edits per step** and an *edit confusion matrix* (edit-precision/recall on corrupted vs clean tokens) (Fig. 1–5).

3) **Fairness/confounds in SOTA comparisons.**
Table 3 mixes distinct tokenizers (e.g., LlamaGen codebook, VAR multi-scale), step counts, and training durations; some rows use cfg and others “No cfg.” Please re-run **matched-tokenizer, matched-compute** baselines (same tokenizer, same training steps, same classifier-free guidance, same image count for FID) and add wall-clock throughput and p95/p99 latency for sampling (see p.xx, Table 3).

4) **Speed claims need breakdown and variance.**
Appendix D states 0.25 s/image on A100 and 2.86× vs vLLM-optimized LlamaGen-XL, but no per-component breakdown (attention vs correction vs I/O) or variability. Provide images/s, latency histograms, and ablate correction disabled/enabled to isolate overhead (Appendix D).

5) **Halton/arccos design is inherited; novelty positioning is weak.**
Halton ordering and convex reveal schedules are known to help MaskGIT-style decoders; here they are adopted from the baseline. Please quantify how much of the gain vs Halton-MaskGIT comes from **random-injection only** (fixing disabled) and how much from **correction** (your contribution). Table 3 hints at +0.87 FID from correction; make this central with controlled runs (Sec. A; Table 6; Table 3).

6) **Distribution shift from “same-image” random injection.**
Eq. (5) injects tokens sampled from the *same image* distribution \(P(Z)\), i.e., realistic but misplaced tokens. Does this bias the model toward spatial relabeling rather than semantic error detection? Please compare against **random-class** and **patch-shuffle** injections and report robustness when corruptions are out-of-distribution (e.g., foreign-object tokens) (Eq. 5; Fig. 5).

7) **Statistics are thin; many gains could be within noise.**
No μ±σ across seeds; Table 2/3 improvements over strong baselines are modest in places. Please add ≥3 seeds (FID50k with CI), paired tests, and per-class precision/recall for ImageNet (Tables 2–3).

8) **Scope is narrow (no text-to-image).**
All image experiments are class-conditional; text-conditional is not evaluated. Since token incompatibilities are severe with text prompts, include at least a small-scale T2I study to establish relevance (Sec. 4).

9) **Video results are not competitive and confounded by tokenizer.**
UCF-101 FVD (242) is behind MAGVIT-family; authors attribute this to OmniTokenizer. Please (i) show BIGFix with **the same tokenizer** as MAGVIT/MAGVIT2, and (ii) report edit statistics on videos (how many tokens corrected per frame) (Table 9; §E).

10) **Reproducibility: configs, seeds, and code.**
Hyper-parameters are summarized (Table 8), but code, exact Halton sequences, α scheduling, and evaluation scripts are needed for verification; release minimal artifacts to reproduce Table 3 on a single GPU (Table 8).

11) **Known limitations affect generality.**
You acknowledge fixed unmasking schedules and <1B-param scale limits. Please quantify quality loss when varying the reveal schedule at test time, and provide a path to larger-scale models (Sec. G).

**Questions:**

See Weaknesses

---

> ### Author Response · Authors · 2025-11-21
>
> **Q1: Correctness of the probabilistic model**
>
> We approximate the joint conditional \(P(z_3,z_4 \mid z_1,z_2)\) by factorizing it into separate conditionals, introducing an error due to the mutual information between \(z_3\) and \(z_4\). To reduce this error, we apply a noising function \(\Phi\) and train the model to denoise/fix tokens, improving the factorized approximation to the true joint distribution. Please see Appendix H for more details.
>
> **Q2: Self-correction mechanism theory.**
>
> All the tokens in the context are eligible to be corrected during sampling. In fact, the model makes a new prediction at inference time over all the tokens in the context from which we resample the new tokens.
> As explained in (Line 377) the model “corrects 58 tokens per image on average.”, but we updated Table 4 during rebuttal. The role of the scheduler is already explained in Appendix A and in Table 5.
>
> **Q3: Fairness in Tab 3**
>
> This is the SOTA table which aims at comparing best related works that achieve the best score against our best model regardless of the type of methods, number for steps, tokenizer. Running with the same tokenizer is indeed not possible as some models use continuous tokenizers (e.g. SiT, DiT). We do not have the compute power to train all the models on all the different tokenizers existing in the SOTA table or their training steps sometimes requiring huge amounts of compute.
>
> **Q4: Speed**
>
> We measured inference speed for each model across multiple decoding steps (the mean over 1000 samples with batch size of 8*2 for CFG). From these runs, we report the steps, the batch latency, per-image latency and the throughput (images/sec). The experiments are made on an Nvidia A100 and bfloat16 precision and CFG. We do not use KV-cache or any other tricks. The correction does not cause any overhead, but would prevent KV-cache using our current implementation. We also add below in new Table 9 the complete table for Inference Speed across model size, input dimension and number of steps.
>
>
> | Model  | Step | Input size 16×16 (img/s) | Input size 24×24 (img/s) |
> |--|--|--|--|
> | **\ours{}-Tiny**   | 8    | 80.63                    | 41.98                    |
> || 16   | 55.89                    | 30.32                    |
> || 24   | 42.53                    | 23.48                    |
> || 32   | 34.45                    | 19.30                    |
> | **\ours{}-Small**  | 8    | 71.66                    | 37.28                    |
> || 16   | 47.18                    | 25.51                    |
> || 24   | 35.30                    | 19.39                    |
> || 32   | 28.21                    | 15.61                    |
> | **\ours{}-Base**   | 8    | 54.03                    | 27.17                    |
> || 16   | 33.39 | 16.87|
> || 24   | 24.16 | 12.21|
> || 32   | 18.88                    | 9.58|
> | **\ours{}-Large**  | 8    | 28.18 | 14.37                    |
> || 16   | 15.53| 7.98|
> || 24   | 10.73| 5.53|
> || 32   | 8.21| 4.22|
> | **\ours{}-XLarge** | 8    | 21.93 | 10.74 |
> || 16   | 11.90| 5.80|
> || 24   | 8.15| 3.97 |
> || 32   | 6.18| 3.01|
>
> **Q5: Novelty**
>
> As stated in the question, the improvement is +0.84 of FID which corresponds to a 25.89% decrease in the FID, which is significant. We kindly ask the reviewer to clarify the question.
>
> **Q6: Distribution shift**
>
> In all our experiments, using \alpha >0 lowers the FID, this suggests that there is no distribution shift.
> “Does this bias the model toward spatial relabeling rather than semantic error detection?” We would appreciate guidance in how the reviewer would like us to evaluate this?
>
> **Q7: Statistics are thin**
>
> We believe that the results are significant and consistent across experiments on multiple datasets, reproducing Table 3 with  more than 3 seeds would be required to train 60 networks which surpasses our current capacity.
>
> **Q8: text-to-image**
>
> We include ablations on CC12M+MidJourneyDB, which offer preliminary insight into the applicability of our approach to text-conditioned generation. A bigger model will be added before the end of the rebuttal.
>
> | Methods| #Param. | Train Steps | Step | Aesthetic | ImageReward | HPSv2 | PickScore |
> |-|--|--|--|--|--|--|--|
> | Ours (α=0.0)|437M|500k|32|5.34|0.17|0.23| 0.210|
> | Ours (α=0.1)|437M|500k|32|5.60|0.39|0.25|0.215|
> | Ours (α=0.2)|437M|500k|32|5.71|0.53|0.25|0.217|
>
> **Q9: Video results**
>
> As explained in Appendix E, MagVIT is a closed-source tokenizer thus we cannot use them. Moreover, our results on Video synthesis shows competitive results while having a limited compute budget (Table 9 and Table 11).
>
> **Q10: Reproducibility** We will release the code for reproducibility upon acceptance of the paper.
>
> **Q11: Larger Scale Models**
>
> We present an estimate of the scaling law in Figure 7b and Table 2, which indicates that the improvements remain consistent as the model size increases. This trend suggests that our method would likely achieve even better FID scores when scaled beyond 1B parameters.

---

### Official Review · Reviewer_chfx · 2025-11-03

**Soundness:** 2
**Presentation:** 2
**Contribution:** 2
**Rating:** 2
**Confidence:** 3

**Summary:**

The paper introduces BIGFix, self-correcting image generation by iteratively refining sampled tokens. It addresses a key limitation in masked generative transformers (e.g., MaskGIT) by injecting random tokens during training to teach the model to recognize and fix structural inconsistencies at inference time. The authors demonstrate that this “token fixing” strategy improves both visual fidelity and robustness while maintaining parallel generation efficiency. Experiments on ImageNet-256, CIFAR-10, UCF-101, and NuScenes show lower FID/FVD scores and faster convergence compared with diffusion, flow-matching, and autoregressive baselines.

**Strengths:**

- Introducing self-correction into multi-token image generation is a meaningful contribution.
- BIGFix could be integrated with existing token-based architectures (e.g., MaskGIT, LlamaGen, OmniTokenizer) without network modification, diffusion steps, or extra fine-tuning stages. This means the proposed method improves quality and robustness without increasing model size or inference cost.
-Provide extensive experiments spanning both images (ImageNet-256, CIFAR-10) and videos (UCF-101, NuScenes), and a systematic ablation study on α (token injection rate), number of steps, scheduling strategy, and token ordering

**Weaknesses:**

### Major:
- It is unclear to me how this work differs from previous works, given that the random token injection resembles existing denoising or noise-robust training schemes. A more formal analysis of why token fixing improves sample consistency would be better.

- The independence assumption Eq1. P(za, zb) \approx P(za)P(zb) is qualitatively discussed but not rigorously explored. There is no analysis of joint token entropy or mutual information to validate the rationale for the correction.

- Scalability seems limited.  Experiments are mostly run with models that have fewer than 1 billion parameters and often reuse open-weight tokenizers. Claims about generality and large-scale lack evidence from larger models or text-to-image tasks.

### Minor:
- No failure case analysis. Failure modes (e.g., over-correction, semantic drift) are not analyzed.

- Comparison to recent Rectified Flow and MaskGIT-v2 results is outdated; the absence of open-weight diffusion transformers (e.g., PixArt-Σ, SD3) limits the context of “state-of-the-art” claims.

### Others
- The writing style of the paper is non-academic, especially in the introduction and conclusion sections. It is recommended to rewrite the introduction in a conventional scientific exposition style to better align with academic requests.

**Questions:**

- The “token fixing” mechanism relies on random token injection during training to encourage self-correction, but the paper does not provide a theoretical justification of why this improves joint token consistency. Please provide an analysis.
- The paper mentions that the model “corrects 58 tokens per image on average.”(Line377) Could the authors quantify the effect of the correction on global FID? For instance, does the number of corrected tokens correlate with perceptual improvement?
- While Random Token Injection α (Line317-32e) controls token corruption during training, does α also affect correction aggressiveness during inference?Would dynamically adjusting α (annealing or adaptive) further improve generation?

---

> ### Author Response · Authors · 2025-11-21
>
> **Q1: Novelty**
>
> Our baselines are multitoken prediction methods where sampling errors typically occur. In previous discrete methods (MaskGIT, VAR, or HaltonMaskGIT), no noise is injected during training. We show that our noise is not sampled randomly from the full codebook but specifically from the image token space, effectively mimicking sampling errors. This approach is novel and, to our knowledge, unexplored in prior work. Furthermore, token fixing is a capability not reported in any related multi-token prediction methods for image synthesis.
>
> **Q2: Independence Assumption**
>
> Multi-token prediction models are trained on the cross-entropy loss for each token individually, and, therefore, only provide an estimate of each token marginal distribution instead of the joint distribution. We refer the reviewers to [1], which provides an in-depth analysis of MI related to this assumption.
>  [1] V. Besnier et al., “Halton Scheduler for Masked Generative Image Transformer,” ICLR 2025
>
> **Q3: Scalability and Generality**
>
> Training models >1B params requires extensive GPU resources that are beyond our current capacity, but this is not a limitation of our method. Nevertheless, Fig. 7b and Tab. 2 show the improvement of our method up to ~700M params when increasing the network capacity which gives insight on how the model can perform at larger scale.
>
> For text-to-image, we train our methods on cc12m + MidjourneyDB. Our conclusion remains similar as on other datasets: injecting noise during training (\alpha>0) and allowing the model to fix errors during sampling consistently improves the sample quality. We will release a bigger model before the end of the rebuttal period.
>
> | Methods            | #Para. | Train Steps | Step | Aesthetic | ImageReward | HPSv2 | PickScore |
> |--------------|-----|-----|---|--|--|--|--|
> | Ours (α=0.0)      | 437M   | 500k        | 32   | 5.34      | 0.17        | 0.23  | 0.210     |
> | Ours (α=0.1)      | 437M   | 500k        | 32   | 5.60      | 0.39        | 0.25  | 0.215     |
> | Ours (α=0.2)      | 437M   | 500k        | 32   | 5.71      | 0.53        | 0.25  | 0.217     |
>
> **Q4: Failure Case**
>
> Sometimes the model over-corrects certain areas, which results in an overly smooth background. We will include some visualizations. Exploring additional correction strategies might help preserve more image details in future work.
>
> **Q5: Missing Baselines**
>
> We do not extensively compare with Diffusion or FM models, as our model works on a discrete latent space. But we already compare our method with SiT (see Table 2 and 3), a RF model. We kindly ask the reviewer to provide the reference for MaskGIT-v2.
> Moreover, in our first version, our work did not address text-to-image tasks thus we did not compare with PixArt-Σ or SD3.
> In the revised version, as suggested by reviewers g51t and QMWq, we added two recent baseline methods in Tab 3.
>  -ZipAR [2], a method that significantly accelerates inference of AR but it typically impact FID (it increases LlamaGen-XL FID from 2.83 to 3.67). Our method both improves FID while requiring only a small number of forward passes.
> - NAR [3], using similar sampling budgets, we obtain better FID (2.58 vs. 2.48), prec. (0.82 vs. 0.83), and recall (0.57 vs. 0.55), though IS is lower (293.5 vs. 252.5). Importantly, our model is significantly smaller than the architecture used in NAR.
>
> [2] ZipAR: Parallel Autoregressive Image Generation through Spatial Locality. ICML 2025
> [3] Neighboring Autoregressive Modeling for Efficient Visual Generation. ICCV 2025
>
> **Q6: Token Fixing Mechanism**
>
> We approximate the joint conditional \(P(z_3,z_4 \mid z_1,z_2)\) by factorizing it into separate conditionals, introducing an error due to the mutual information between \(z_3\) and \(z_4\). To reduce this error, we apply a noising function \(\Phi\) and train the model to denoise/fix tokens, improving the factorized approximation to the true joint distribution. See new Appendix H for more details.
>
> **Q7: Impact of Token Corrections on FID**
>
> The number of corrected tokens does not necessarily correlate with FID. Consistent with our conclusion in l.317–332, increasing α leads the model to correct more tokens during inference (new Table 4). This initially improves FID, but beyond a certain point the model begins to over-correct, producing overly smooth images and reducing diversity, reflected by the increase in both FID and IS.
>
> | Steps| α|Value|
> |--|--|--|
> | **8**|0.1|4.47 |
> ||0.2| 15.91 |
> ||0.3| 38.15 |
> |**16**| 0.1| 7.22|
> ||0.2| 22.69|
> ||0.3| 50.14|
> | **24**| 0.1|8.77|
> || 0.2|26.79|
> || 0.3|58.21|
> |**32**| 0.1|10.23|
> || 0.2|30.47|
> || 0.3|65.41|
>
> **Q8: α on Inference-Time**
>
> We would like to emphasize that the parameter α, is used only during training; no noise is injected for sampling. However, increasing α during training leads the model to perform more aggressive token corrections at inference, as shown in the table above. Furthermore, dynamically adjusting α did not yield improvement.

---

### Author Response · Authors · 2025-11-21
**Final Version**

We thank all reviewers for their detailed feedback, and acknowledgement for the paper's clear problem framing and effective token-fixing mechanism, showing strong results across image and video benchmarks. During the rebuttal, we have strengthened the submission with new experiments, additional baselines, clarified methodology, new tables, figures, and theoretical explanations. Below we highlight the major additions and clarifications, and we leave individual comments for specific questions.

**(A) New baselines and fairer comparisons**

We added three Halton-MaskGIT models (S/B/L) trained for 410k steps, using their official implementation, strengthening the direct comparison with our variants at matched compute. This now appears in **revised Table 2**, showing that BIGFix improves significantly over strong Halton scheduling baselines. We also added comparisons with the two parallel/multitoken prediction methods highlighted by reviewers: ZipAR and NAR, both are now discussed in the Related Work (L174–177) and added as baselines in **Table 3**.

**(B) New text-to-image experiments (CC12M + MidJourneyDB)**

Although large-scale T2I training (PixArt, SD3, FLUX) is not feasible within the rebuttal time window, we performed a 7M-image text-to-image experiment for 500k iterations with 437M parameters. Results (**new Table 4**) show consistent improvements for α > 0, confirming generalizability. We also train a 627M-parameter model and fine-tune (50k train steps) it at higher resolution on Laion Aesthetics and Diffusion 4K datasets. With only 32 sampling steps, our method achieves the best performance for models below 1B parameters (see **new Table 5** and **new Figures 7 and 8**).

**(C) New inference-speed analysis with full tables**

We added a complete inference speed table (**new Table 13**) covering 5 model sizes, 4 decoding-step budgets (8/16/24/32), and two resolutions (16×16 and 24×24 latents). These results address all practical speed questions and demonstrate that our correction mechanism does not add overhead for inference.

**(D) Quantitative analysis of token-correction behavior**

We added the full table of average tokens corrected per image (Steps × α) in the **revised Table 8**. We also clarify that α is used only during training, not sampling, where we correct errors naturally arising due to multi-token prediction paradigms.

**(E) Scaling-law evidence**

Figure 9b and **updated Table 2** demonstrate that improvements scale consistently with model size, supporting the feasibility of models with >1B parameters.

**(F) Theoretical grounding of the Fixing correction**

We added a paragraph in **appendix A** to clarify the role of the correction mechanism and explain how the noise-injection and fixing steps lead to a better approximation of the true joint distribution.

---

> ### Author Response · Authors · 2025-12-03
>
> We edited our previous comment and uploaded the revised paper, with all changes from the original submission highlighted in blue.

---

### Meta-Review · Area_Chair_9L93 · 2026-01-11

**Summary:**

Reviewers agreed the paper targets a real pain point in parallel / multi-token prediction for visual generation (structural inconsistencies) and found the “token fixing” idea intuitively appealing. However, the recommendation is driven by concerns
1. the core novelty and probabilistic justification are not clearly established beyond noise-injection/denoising-style training,
2. comparisons and attribution are confounded like mixed tokenizers/training setups, unclear isolation of where gains come from, and
3. the evidence for generality/scaling like text-to-image, larger models, strong video baselines is still limited.

Overall, reviewers did not find the current submission sufficiently positioned for acceptance.

**Reviewer Concerns:**

Addressed by the rebuttal (partially):
1. added stronger/more relevant baselines (Halton-MaskGIT at matched compute; ZipAR/NAR) and expanded comparison tables
2. added preliminary text-to-image results and more inference-speed reporting
3. added additional explanation/theory (MI-based motivation) and quantitative tables on token correction behavior

Still outstanding:
1. Novelty + theory: reviewers asking for a clearer, more formal explanation of why the training objective yields targeted correction, and how it differs from existing noise-robust/denoising-style training) remain only partially satisfied
2. Clean attribution + fairness: key confounds remain like tokenizer/training differences; unclear controlled separation of gains from Halton ordering vs token injection vs fixing; limited statistics/no multi-seed CIs
3. Generality: text-to-image and larger-scale claims are only supported by limited mid-scale experiments; video comparisons remain less convincing and are tokenizer-dependent
4. Clarity + reproducibility: algorithmic description of “fixing” is still not as crisp as reviewers requested, such as pseudocode, error/overwrite analysis

**Reviewer Scores:**

YYtR (8): likely unchanged.
QMWq (4): may bump up a bit due to added baselines + speed table + preliminary T2I evidence, but still concerns on clarity/positioning.
g51t (4): likely unchanged. novelty concerns persist
chfx (2): likely unchanged. core theory/novelty/scaling concerns remain
LeAz (2): likely unchanged. even discounting review quality, similar concerns are echoed elsewhere

---

### Decision · Program_Chairs · 2026-01-26

Reject